# Photosynthetic Light-Harvesting (Antenna) Complexes—Structures and Functions

**DOI:** 10.3390/molecules26113378

**Published:** 2021-06-03

**Authors:** Heiko Lokstein, Gernot Renger, Jan P. Götze

**Affiliations:** 1Department of Chemical Physics and Optics, Charles University, Ke Karlovu 3, 12116 Prague, Czech Republic; 2Max-Volmer-Laboratorium, Technische Universität Berlin, Straße des 17. Juni 135, D-10623 Berlin, Germany; 3Institut für Chemie und Biochemie, Freie Universität Berlin, Arnimallee 22, D-14195 Berlin, Germany; jgoetze@zedat.fu-berlin.de

**Keywords:** bacteriochlorophylls, chlorophylls, carotenoids, excitation energy transfer, light-harvesting complexes, photosynthesis, pigment-protein complexes, photosystems, photoprotection

## Abstract

Chlorophylls and bacteriochlorophylls, together with carotenoids, serve, noncovalently bound to specific apoproteins, as principal light-harvesting and energy-transforming pigments in photosynthetic organisms. In recent years, enormous progress has been achieved in the elucidation of structures and functions of light-harvesting (antenna) complexes, photosynthetic reaction centers and even entire photosystems. It is becoming increasingly clear that light-harvesting complexes not only serve to enlarge the absorption cross sections of the respective reaction centers but are vitally important in short- and long-term adaptation of the photosynthetic apparatus and regulation of the energy-transforming processes in response to external and internal conditions. Thus, the wide variety of structural diversity in photosynthetic antenna “designs” becomes conceivable. It is, however, common for LHCs to form trimeric (or multiples thereof) structures. We propose a simple, tentative explanation of the trimer issue, based on the 2D world created by photosynthetic membrane systems.

## 1. Introduction

(Bacterio)chlorophylls, (B)Chls, being noncovalently bound to specific apoproteins, are the major chromophores of most light-harvesting (antenna) complexes (LHCs). Moreover, together with metal-free (bacterio)pheophytins, (B)Pheo, (B)Chls are the principle photochemically active pigments in photosynthetic organisms. Excitation arriving at the lowest singlet excited state (S_1_) of special (B)Chls in the photochemical reaction centers (RCs) eventually leads to primary charge separation, thus converting light energy into electrochemical energy, ultimately providing the driving force for all processes in photosynthetic (and also heterotrophic) organisms [1]. The vast majority of (B)Chls, however, act as (accessory) light-harvesting pigments, increasing the optical cross sections of the photochemical RCs by about two orders of magnitude and more [2].

(B)Chls are essential constituents of the RCs and account for the majority of the antenna pigments in nonoxygen-evolving (anoxygenic) photosynthetic bacteria. In all oxygen-evolving photosynthetic organisms, Chls *a* functions as a primary electron donor in the RCs of both photosystems I and II (PSI and PSII). Moreover, a Chl *a* is also the primary electron acceptor in PSI. An exception may be the cyanobacterium *Acaryochloris (A.) marina*, in which Chl *d* also appears to be involved in electron transfer [3]. In higher plants and green algae, Chls *a* and *b* serve as the major antenna pigments in the LHCs. Chl *b* is found only in the LHCs of higher plants and green algae, with most Chl *b* bound to the main light-harvesting complex (LHCII). Chls *a* and *b*, and in some species also divinyl-Chls *a* and *b*, are bound to the “prochlorophyte” (certain clades of the cyanobacterial radiation now better termed oxychlorobacteria) Chl-binding antenna complexes (Pcbs) [4]. Chls *c* are found in LHCs of different algal clades [5]. Chl *d* and the recently discovered Chl *f* [6] are found only in certain cyanobacteria. Chl *d* is the dominant pigment in *A. marina* [7,8]. The considerably redshifted-absorbing Chl *f* was first discovered in stromatolite-forming cyanobacteria [6]. The synthesis of Chl *f* and alternative Chl-binding proteins seems to be a specific response of certain cyanobacteria to grow under far-red light conditions [9]. Whether Chl *f* is also involved in electron transfer, or only serves as an accessory light-harvesting pigment, is currently debated (e.g., [10,11]).

In addition to (B)Chls, most photosynthetic LHCs also bind carotenoids. In plants and algae, LHCs exclusively bind xanthophylls (oxygen-containing derivatives of carotenes). Bacterial LHCs and cyanobacterial, algal and plant core-antenna complexes of both PSI and II can contain both carotenes and their oxygen derivatives. The functions and photophysics of carotenoids (also when bound to LHCs) have been covered extensively in a previous review [12]. Carotenoids have unique photophysical features that render their spectroscopic investigation, in particular in LHCs, rather difficult. Conventionally, carotenoid excited states are assigned due to an assumed C2h symmetry (in analogy to linear polyenes). However, very recent studies have suggested that the underlying assumptions may not apply to carotenoids, in particular to the optically “dark” first excited singlet state (S_1_) [13].

Structures of Chl *a* and violaxanthin (as representatives of their molecule classes), as well as the corresponding absorption spectra and energy levels, are shown in Figure 1. In spite of the vast taxonomic diversity of photosynthetic organisms, the mechanisms of the primary photochemical reactions performed by (B)Chls and the other associated cofactors in the RCs are very similar, while—in marked contrast—both the structures of LHCs and the bound pigments exhibit a remarkable variety.

This article reviews the functional similarities and structural diversity of photosynthetic LHCs. We will also explore how computational modeling of LHC functions benefits from the availability of high-resolution structures.

## 2. The Photosynthetic Apparatus of Plants (and Algae)

Photosynthetic electron transport in all oxygenic photosynthetic organisms is driven by two light reactions, each taking place in two membrane-integral multimeric pigment-protein super complexes referred to as PSI and PSII. Higher plant thylakoid membranes are characterized by distinct regions: (i) the loose lamellar stroma thylakoids and (ii) the tightly stacked grana. The grana stacks are highly enriched in PSII. PSI super complexes are mainly found in the stroma-thylakoid regions [14]. These structures are, however, highly dynamic and undergo changes on different time scales, see, e.g., [15].

Pigment-protein complexes acting as LHCs absorb photons and effectively transfer the electronic excitation energy in a series of ultrafast steps (on the order of less than 100 femtoseconds up to several picoseconds [16]), with the excitons eventually arriving at the RCs.

Two basic modes of excitation energy transfer (EET) can be distinguished: in the case of weak coupling of the interacting pigments due to distance or less favorable orientation, EET occurs as an incoherent “hopping”-like Förster-type transfer. However, if the chromophores are in close contact and in favorable orientation, strong excitonic coupling may result. The interacting molecules will share their electronic ground and excited state manifolds, and splitting (in case of identical chromophores) or shifting of the resultant excited state energy levels may occur. The energies and oscillator strengths of the coupled excited states can be used to determine the orientation of the involved transition dipole moments (or vice versa, in case of computational predictions) since the oscillator strengths of the coupled states are redistributed according to the orientation of the involved transition dipole moments and the orientation of the coupled chromophores. Excitations may be delocalized among the strongly coupled pigment molecules. Interestingly, both modes of pigment-pigment coupling occur in most LHCs.

(B)Chls of RCs, usually characterized by redshifted, i.e., lower energy absorption maxima as compared to antenna pigments, act as traps for the excitation energy and are the photochemically active pigments in the subsequent charge separation processes (cf. [17]). Based on structural and functional similarities of the RCs of anoxygenic photosynthetic bacteria, “special pairs” of Chls were inferred to act as primary electron donors (termed according to their specific absorption maxima, P680 in PSII or P700 in PSI). This conventional view has to be revised due to recent spectroscopic studies [18,19,20,21]. In fact, four Chl *a* molecules in PSII appear to be strongly excitonically coupled, with the lowest exciton state being predominantly localized on the monomeric Chl_D1_, which probably functions as the primary electron donor to Pheo_D1_ as the acceptor. This step is followed by rapid electron transfer from the “special pair” P_D1_P_D2_ to Chl 
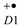
[22]. Alternative and complementary pathways are being discussed for the primary charge separation in PSII [23]. Analogously, also in PSI, Chl *a* was shown to act as the primary electron donor, which transfers an electron onto another Chl *a* (denoted A_0_) as the acceptor. Subsequently, the “special pair” P700 (actually being a *hetero*dimer of Chl *a* and Chl *a*’, its C13 epimer) is rapidly oxidized [19,24].

Electron transport from PSII to PSI is coupled with proton translocation from the stroma into the thylakoid inner (luminal) space which gives rise to a difference of the proton electrochemical potential across the membrane. The resulting proton motive force is used to synthesize ATP [25,26]; for a review, see, e.g., [27]. Moreover, the acidification of the thylakoidal lumen, i.e., ΔpH formation, is also important for the regulation of light harvesting in PSII and interphotosystem electron transport (see below). 

Chloroplasts are semiautonomous organelles, i.e., they contain genetic information in their plastome. The plastome encodes for some core polypeptides of PSI (~5) and PSII (~14), particularly those binding cofactors for light-induced charge separation. The majority of the up to 3000 thylakoid proteins, including the LHCs, are, however, encoded by nuclear genes [28].

### 2.1. Plant (and Algal) Peripheral Light-Harvesting Complexes

In plants and green algae, the peripheral antenna systems of both PSI and PSII are made up from a large family of light-harvesting Chl *a/b*-binding LHC proteins encoded by the *lhca* and *lhcb* gene families, respectively. Three major types of peripheral Chl *a/b*-binding LHCs can be distinguished in higher plants [28].

LHCII is the main antenna complex comprising about half of the thylakoid protein and binding roughly 50% of the Chls. LHCII forms homo- and heterotrimers of three members of the protein superfamily, Lhcb1, Lhcb2 and Lhcb3. LHCII can also be associated with PSI. This would be the case, in particular, following so-called state 1-state 2 transitions [29].

The monomeric minor LHCs of PSII are formed by the Lhcb4–6 proteins (also termed CP29, CP26 and CP24, respectively, according to their apparent molecular weights), with one copy of each located proximal to the PSII core complex. Recently, structures of PSII mega complexes, including a part of the more tightly associated proximal LHCs, have become available [30].

The peripheral antenna proteins of PSI are designated Lhca1–4. One copy of each of these proteins is arranged in a single layer around the PSI core under all physiological conditions. Lhca1 and Lhca4 apparently form a heterodimer, which gives rise to the characteristic long-wavelength fluorescence emission of plant PSI at low temperatures [31,32,33].

The structure of LHCII (Figure 2) has been obtained to atomic (2.72 and 2.5 Å) resolution by X-ray crystallography [34,35].

The LHCII monomeric subunit is thought to provide a general structural model for all Chl *a/b*-binding proteins and the Chl *a/c*-binding proteins of certain chromophyte algae, as well as the PSI-LHCs of rhodophytes, due to substantial protein sequence homologies. Homologous LHC proteins from different plant species exhibit high (up to 90%) similarity, in particular among angiosperms. Differences among the various members of this protein family can be more substantial (up to 65% identity). All LHCs feature three membrane-spanning α-helices, see Figure 2 and Figure 3. The evolutionary-related first and the third helices and their N-terminal regions exhibit particularly significant homologies.

Fourteen Chl molecules (eight Chls *a* and six Chls *b*) and four xanthophylls (two luteins, one neoxanthin and one xanthophyll-cycle pigment; either violaxanthin, antheraxanthin or zeaxanthin), are noncovalently bound to each monomeric subunit of LHCII, see Figure 2 [34,35]. The Chl molecules are positioned to optimize EET within one subunit as well as between the complexes [37]. Their chlorin rings are arranged roughly perpendicular to the membrane plane in two layers, close to each surface of the membrane. Some Chl *a* molecules are in intimate contact with xanthophylls providing photoprotection, see also below [35,38]. EET between LHCII trimers is facilitated by strongly excitonically coupled Chls, constituting the so-called long(est) wavelength-absorbing terminal emitter(s) at the periphery [37], thus facilitating rapid migration of excitons in the antenna systems(s). 

These observations are in agreement with in vitro reconstitution studies with heterologously overexpressed mutant LHCII-apoproteins, in which putative Chl-binding amino acids had been replaced by nonbinding ones (e.g., Rogl et al. [39]). The latter study suggested that a Chl *a* (termed Chl a2 in the nomenclature of Kühlbrandt et al. [38] or Chl*a* 612 according to Liu et al. [34]) is strongly coupled to a nearby Chl (originally tentatively assigned to Chl “b2”), apparently forming the “terminal emitter” or lowest energy state at about 680 nm in the LHCII EET chain. The higher-resolution X-ray structural analyses of spinach and pea LHCII [34,35] show that Chl “b2” is actually also a Chl *a* (termed Chl *a* 611 by Liu et al. [34]). However, a nearby Chl *b* is apparently closely coupled to this cluster [40]. At low temperatures (77 K and below), however, the lowest state may be associated with other Chls, as suggested earlier [38]. This is also corroborated by a recent re-evaluation of spectral hole-burning data [41].

The degree of (macro-)aggregation of LHCII may vary in vitro as well as in planta. Aggregation strongly determines the photophysical properties of LHCII, i.e., excited state dynamics, including ultra-fast EET and excitonic interactions between certain individual chromophores [42,43].

The structure of the minor PSII antenna complex CP29 (Figure 3) has been solved by X-ray crystallography [36]. This achievement revealed marked differences to structural models derived on the basis of the LHCII structure and two-dimensional electronic spectroscopy, 2DES [44]. In particular, the number of pigments bound to CP29 significantly deviates from previous estimates. The recent structure shows that 13 Chls are bound (8 Chls *a* and 4 Chls *b;* a tentative “mixed site” is also assumed to exist, but not supported by other data). An ab initio computational study provides insight into the optical properties and EET processes in CP29 [45].

Thus far, much theoretical research on LHCs has been focused on quantification of site energies and excitonic couplings of the chromophores [46,47], as well as determination of the molecular mechanism(s) of photoprotection. Computational studies of photosynthetic pigment-protein complexes strongly rely on the (crystal) structures and subsequent “ensemble” generation using classical dynamics. However, it was shown that the chromophore geometry still requires careful treatment before use for in silico-excited state calculations [48]. Especially for the carotenoids, these features need to be computed with appropriate methods [49]. Moreover, due to the size of the involved complexes, most high-accuracy studies focusing on structure/spectrum relationships employ gas-phase model systems of isolated chromophores and/or chromophore pairs. A comprehensive review of these studies is beyond the scope of the current work. However, several recent reviews cover a large portion of the computational research on photosynthetic pigment-protein assemblies, especially those including the protein matrix [50,51,52,53]. Moreover, a more general “bottom-up” perspective of photosynthetic light-harvesting has also been published [54].

More recent work, however, has gone beyond those limitations, introducing full-scale pigment couplings constructing excitonic states from the properties of the individual chromophores [55]. These studies are assumed to shape the future of the field: the full dynamics of EET can only be assessed by a more holistic approach, including all components of the system, despite the necessarily involved (computational) costs.

### 2.2. Core Antenna Complexes

The PSII of all oxygenic photosynthetic organisms contains two core antenna complexes (Figure 4), designated as CP43 and CP47, due to their apparent molecular weights. These proteins are members of an extended protein family [4]. All members of this protein family are characterized by six membrane-spanning α-helical domains (Figure 4). Furthermore, both CP43 and CP47 exhibit large loops at the lumenal side, which are essential for the stabilization of a functional water oxidizing complex (WOC) [56]. CP43 apparently binds 13 Chls *a* and ~3–5 β-carotenes, and CP47 binds 16 Chls *a* and ~4 β-carotenes. The D1 and D2 polypeptides, each forming five transmembrane helices, provide the matrix for noncovalent binding of all cofactors related to light induced charge separation and electron transport in the proper RC of PSII. Thus, the complex CP43/D1/D2/CP47 has a total of 22 membrane-spanning helices (cf. Figure 4).

The situation is different in PSI, where both antenna pigments and all cofactors for electron transport are bound to a heterodimer of the polypeptides PsaA and PsaB, each having 11 membrane spanning helices (Figure 5). In PSI, the core antenna(e) can be considered to be “fused” to the RC. A monomer of the PSI core binds 96 Chls *a* (and 22 carotenoids) [58]. Previously it was assumed that all the carotenoids in the cores of PSI and PSII are β-carotenes; however, recent data indicate that other carotenoids (the xanthophylls zeaxanthin, canthaxanthin, echinenone) also occur, possibly with specific functions [59]. Interestingly, carotenoids in PSI are close to clusters of Chls *a* which may constitute as the so called “red Chls”, due to their redshifted absorption as compared to P700 (some of which have been assigned to specific sites in the structure, see, e.g., [60]).

### 2.3. Peridinin-Chlorophyll a-Protein (PCP)

Another peripheral light-harvesting complex—which is not even remotely evolutionary nor structurally related to the extended LHC superfamily—is the water-soluble (not membrane integral) peridinin-Chl *a*-protein (PCP) found in dinoflagellates. PCP is unique among all known antenna complexes: it uses peridinin (an unusual, highly modified carotenoid) as its main light-harvesting pigment. An X-ray structure was obtained for PCP from *Amphidinium carterae* [62] (Figure 6). Interestingly, PCP also forms trimers in vivo. Each monomeric subunit of the PCP from *Amphidinium carterae* binds eight peridinins per two Chls *a* [62].

Within each monomeric PCP subunit, pigments are organized in two clusters related by a pseudo-twofold symmetry (Figure 6). Each of these clusters contains one Chl *a* surrounded by four peridinins within Van der Waals distance. The closest distances between peridinins and Chls *a*, as well as between peridinins in one cluster, are about 4 Å. The distance between the two Chls *a* in a PCP monomer is rather large (about 17 Å). Thus, excitonic interactions between the two Chls in one subunit appear to be negligible in PCP. However, pronounced peridinin–peridinin as well as peridinin–Chl *a* interactions occur [63].

Amino acid sequences of the C- and N-terminal regions of the PCP apoprotein are ~56% identical [62]. Hence, it can be presumed that the site energies of the attached chromophores (peridinins as well as Chls *a*) in the two regions may differ. The arrangement of pigments in PCP renders the complex suitable to disentangle various pigment–pigment interactions (cf. also [63,64]). Peridinins are assumed not to contribute to absorption in the Chl *a* Q*_y_* region in PCP. However, the energetic location as well as the dipole strength of the S_0_ → S_1_ (1^1^A_g_^−^ → 2^1^A_g_^−^) transition of peridinin—in particular in its protein environment in PCP- is still a matter of controversy [63,65,66,67,68,69,70]. 

Interpretation of the carotenoid photophysical properties requires an understanding of the nature of the “optically dark” excited singlet state (S_1_), which has also been addressed in a critical review covering in vitro and in vivo phenomena (and a discussion of the various experimental approaches to detect them) [71].

Two sub-bands (related to the two Chls in one PCP monomer) have been partly resolved in the Q*_y_* band at cryogenic temperatures [72,73]. However, both studies state that the two Chls may be spectrally indistinguishable at RT. 

Nonlinear polarization spectroscopy in the frequency domain (NLPF) was used to disentangle pigment–pigment interactions in PCP at RT [63]. NLPF is a nonlinear four-wave mixing laser-spectroscopic technique (thus relying on similar principles as 2DES, but predating it) that also has a quadratic excitation intensity dependence. Saturation effects can be used to detect minor spectral components. NLPF spectra showed a remarkable substructure in the peridinin 1 1B_u_^+^ absorption region as well as direct evidence for energetic nonequivalence of the two Chls that is not obvious in the RT absorption spectra. The NLPF spectra also suggested that one of the Chls interacts with the “dark” S_1_ (2^1^A_g_^−^)/ICT state of certain peridinins, and/or the S_1_/ICT state must have a small, but non-negligible, transition dipole moment in the 650 and/or 670 nm region [63]. Single-molecule spectroscopy studies with native and reconstituted PCP further corroborated these results [74].

A significant blueshift of the NLPF spectra of PCP at high pump beam intensities can obviously not be explained by the abovementioned energetic nonequivalence of the Chls *a*. An emerging sub-band centered at ~660 nm was assigned to a low-dipole moment S_1_/ICT state of peridinin molecules with a blueshifted 1^1^B_u_^+^ state (absorbing at ~495 nm). 

Dinoflagellates also contain membrane-intrinsic antenna complexes belonging to the LHC family, e.g., the Chl *a*-Chl *c_2_*-peridinin protein complex (apcPC). Currently, no high-resolution structure of these complexes is available. However, these complexes are presumed to be structurally and functionally, also in terms of photoprotection, similar to LHCII [75].

Computational studies on peridinin and PCP also result in similar explanations for EET mechanisms [76,77]. Namely, the relaxation of the allowed 1^1^B_u_^+^ state of peridinin into S_1_ (2^1^A_g_^−^) is not trivial, likely resulting in a mixed state of at least partial ICT character. Depending on the computational methods used, this mixed state may even be initially populated via direct absorption close to the typical xanthophyll allowed state. From this state, which apparently cannot be labeled as pure S_2_ or S_1_, Förster-type EET to the closest Chls *a* may occur. These results, however, are very sensitive to the employed computational approach, since the lowest excited state of carotenoids is very difficult to assess [78]. Still, experimental evidence for the theoretically modelled type of mixed EET (with S_2_ preference) exists [79].

## 3. Antenna Systems of Photosynthetic Prokaryotes

Two fundamentally different types of photosynthetic prokaryotes can be distinguished: anoxygenic phototrophic bacteria, containing just one photosystem of either PSI or PSII type [80] and the oxygenic cyanobacteria. Cyanobacteria contain both PSI and PSII [81] and are assumed to be the evolutionary precursors of chloroplasts in plants (and algae).

### 3.1. Purple Bacterial Antenna Complexes

Photosynthetic purple bacteria are among the best studied phototrophic organisms [82,83,84,85]. Much insight into the mechanisms of the primary reactions of photosynthesis has been gained using purple bacteria as model systems (cf. [83,85]). 

The core and peripheral antenna complexes (designated LH1 and LH2, respectively) of purple bacteria are membrane-intrinsic oligomeric ring-shaped structures consisting of small one-helix α and β subunits, forming heterodimers as basic building blocks [82]. These subunits bind (B)Chl *a* or *b* and carotenoids [85]. The LH1 ring directly surrounds the RCs [86] (cf. Figure 7), while LH2 forms separate, adjacent entities. LH1 features just one (B)Chl ring, giving rise to a single Q*_y_* absorption band at ~875 nm for (B)Chl *a*-containing species or up to 1050 nm for (B)Chl *b*-containing ones.

The number of LH2 per RC may vary, and they are not found in all purple bacteria under all conditions. The structural and functional properties have been described in great detail, e.g., [82,84] and are thus considered only briefly in this review.

The (peripheral) antennae, LH2, comprise of two distinct circular (B)Chl *a* aggregates bound to the apoproteins (Figure 8), termed (B)Chl-B800 and (B)Chl-B850, according to their Q*_y_* absorption maxima (reviewed in, e.g., [64]).

Nonlinear absorption spectroscopy and NLPF of LH2 from *Rhodobacter sphaeroides* has suggested that the (B)Chls *a* of the B850 ring are strongly excitonically coupled and that the excitation energy is delocalized over 16 ± 4 (B)Chl molecules [88,89]. Moreover, NLPF also indicated that an upper excitonic component of (B)Chl-B850 is located in the B800 absorption region [90]. Notably, certain sulfur purple bacterial species (e.g., *Allochromatium minutissimum*) form relatively stable LH2 complexes even in the absence of carotenoids. The comparison of carotenoid-less LH2 with native carotenoid-containing LH2 can provide unique insight into the functions of carotenoids in light harvesting [40,91,92,93].

Due to the highly symmetric arrangements of apoproteins and pigments in LH2, it is a very attractive target for modeling, although the overall size of the system calls for simplifications. A recent theoretical study was able to find specific properties of vibrational modes to be involved in B800/B800 and B800/B850 EET processes [94]. It is also subject of a series of publications concerning both the development and application of a polarizable QM/MM approach, investigating the nature of the involved transitions and couplings and the role of CT states [95,96,97].

### 3.2. Chlorosomes and Other Antenna System Components of Photosynthetic Green Bacteria

A very peculiar construction principle for LHCs is used in the chlorosomes—the peripheral antenna complexes of photosynthetic green sulfur bacteria and green filamentous bacteria, which are not phylogenetically related. Chlorosomes are very large assemblies of well-ordered rod-like aggregates of several 10.000 (B)Chl *c*, *d* or *e* molecules [98]. Only a very few small proteins of yet unknown function are peripherally associated with chlorosomes. Indeed, chlorosomes devoid of these proteins or artificial aggregates of (B)Chls exhibit very similar structural arrangements and spectroscopic features [99].

Chlorosomes are connected to the RCs by the baseplate proteins [100] and the Fenna–Matthews–Olson (FMO) protein [101,102,103]. The FMO protein was the first (B)Chl-binding protein of which a high-resolution X-ray structure was obtained [101,103] (Figure 9). Intriguingly, the FMO protein, too, forms trimers in vivo. FMO binds seven or eight (B)Chls *a* per monomeric subunit [104].

Despite the chlorosomal baseplate being structurally highly ordered, the inner regions of the chlorosome prevent proper atomistic modeling. Few theoretical studies are available [105]. However, due to the well-known structure of FMO, detailed theoretical analyses were performed on the mode of EET in the FMO complex [106,107]. FMO has actually turned into some test and validation systems for various computational approaches [108,109,110]. These approaches are complemented by advanced experimental techniques: recently, the EET flow through the entire photosynthetic apparatus of the green sulfur bacterium *Chlorobaculum tepidum* was monitored using 2DES, integrating previous observations with isolated antenna complexes into a holistic picture [111].

### 3.3. Cyanobacterial Antenna Systems

Cyanobacteria are generally considered to be to be the evolutionary ancestors of plant and algal chloroplasts; however, they do not possess the LHC-type protein (complexes). Cyanobacteria, however, contain functionally analogous peripheral antenna complexes. These employ very different structural construction principles. Instead of membrane-integral LHCs, water-soluble (membrane-extrinsic) phycobilisomes are the typical peripheral antenna systems of cyanobacteria and red algae and also occur in the appropriately named, rather obscure clade cryptophyta [112,113,114]. Phycobilisomes are composed of multiple phycobiliproteins that form rod-like structures and contain phycobilin chromophores (open-chain tetrapyrroles related to bile and phytochrome pigments). In contrast to Chls and carotenoids in other LHCs, phycobilins are *covalently* bound (via thio-ether bonds) to the respective apoproteins. Binding constrains the phycobilin chromophores in an extended (linear) conformation and thus strongly modulates their photophysical properties, i.e., their excited-state energies are shifted and absorption strengths and lifetimes are increased, in particular those of the lowest singlet excited state. Hence, the photophysical properties of phycobiliproteins are not only determined by the type of the bound chromophores—phycoerythrobilin, phycocyanobilin or phycourobilin (possessing strong absorption bands between 450 and 670 nm)—but decisively so also by covalent binding to the apoproteins (Figure 10 and Figure 11).

Certain cyanobacteria can adapt their light-harvesting antenna system to changing ambient light qualities by altering the phycoerythrin to phycocyanin ratio—a phenomenon which is referred to as complementary chromatic adaptation [116].

Interestingly, cyanobacteria possess membrane-spanning one and two α-helix-forming proteins (so called high-light inducible proteins, HLIPs) [117,118,119], of not yet fully established functions, with considerable sequence homology to the α-helical stretches of typical plant LHC proteins (cf. Figure 2 and Figure 3). Moreover, related one-helix proteins (OHPs) also occur in higher plants. OHPs seem to be involved in the assembly of the photosynthetic apparatus of plants and protect Chls from photooxidation [120].

The HLIPs are assumed to be phylogenetic precursors of the LHC superfamily (including plant early light-inducible proteins, ELIPs, and the four-helix PsbS protein, which appears to be critically involved in photoprotection/NPQ, see below). All these LHC and LHC-like proteins are likely to have evolved by multiple gene duplication, fusion and, possibly, helix-deletion events [118,119]. At present it is not clear why the evolution of typical LHCs did not occur in cyanobacteria and started only later in the evolution of rhodophytes, algae and plants.

### 3.4. IsiA and Pcbs

Under stress conditions (in particular, iron deficiency, but also high-light and/or oxidative stress, etc.), cyanobacteria accumulate an additional Chl *a*- (and ß-carotene-) binding protein—**i**ron **s**tress **i**nduced protein **A**, IsiA [121,122]. The IsiA protein, also called CP43′, is structurally (and evolutionary) related to the PSII core antenna complex CP43 (compare Figure 4). IsiA was shown to form octadecameric rings around trimeric PSI [123]. Aggregates of the IsiA protein have been proposed to quench excess excitation energy under light stress conditions [124,125].

Antenna proteins homologous to CP43 (and isiA) are constitutionally expressed in Oxychlorobacteria (previously called “Prochlorophytes”)—(divinyl-) Chl *b*-containing clades of the cyanobacterial radiation. These proteins are consequently called “**p**rochlorophyte **c**hlorophyll *a/b*-**b**inding proteins” (Pcbs). Pcbs can—similar to IsiA—form octadecameric rings around trimeric PSI [126]. Additionally, specific Pcbs were shown to act as PSII peripheral antenna complexes [127].

## 4. In Vitro Reconstitution of LHCs

LHCII and other related antenna complexes can be reconstituted from their components (apoproteins, pigments and lipids) in vitro. In vitro reconstitution studies have been carried out with isolated native [128] or recombinant LHC apoproteins (e.g., [39,129,130]) and have provided a wealth of insight into the requirements for stable assembly of these complexes as well as structure-function relationships. Experiments carried out with native and mutant (truncated) apoproteins, or with different compositions and ratios of cofactors, have revealed the requirements of protein structural domains and cofactors that are indispensable for the reconstitution of stable LHCII [39,131]. Proteins, Chls, specific xanthophylls and specific lipids altogether are required for the stability of the LHCs in vitro and in vivo. Lutein has been shown to be required for LHCII-trimer formation in vitro and in vivo [130,132]. The lipid phosphatidyl glycerol (PG) is required for LHCII-trimer formation as well [130]. The presence of another lipid, digalactosyl diacylglycerol (DGDG), is essential for macroaggregate formation and, consequently, also for crystallization of trimeric LHCII [130]. Pigments and apoproteins are degraded in vivo if one of these components is missing. PG apparently also coordinates a Chl *a* molecule which is likely to be involved in EET from LHCII to CP29. The same Chl *a* is also assumed to be participating in a recently proposed excitation energy quenching mechanism [133,134].

## 5. Photodamage, Photoprotection and Regulation of Functional Antenna Size

The essential functional role of antenna systems is the optimal adaptation to quite different environmental—mainly illumination—conditions that may vary, both in time (seasonal, diurnal, even in fractions of seconds) and space (e.g., canopy region versus ground level in a tropical rain forest or different depths in oceans and lakes).

A key parameter of the regulatory control is the adjustment of functional antenna size. The response to variations of environmental—in particular, light—conditions comprises different time scales: high-light stress triggers a short-term response (within seconds to few minutes) leading to the alteration of light-harvesting efficiency in the PSII antenna system via enhanced nonradiative dissipation. This mechanism apparently involves the action of the so-called xanthophyll cycle (this important photoprotective mechanism is discussed in more detail below). The so-called state 1-state 2 transitions regulate the balance of excitation energy distribution between both photosystems via lateral diffusion of LHCII (in particular, under low light conditions, cf., e.g., [135,136,137]). State 1-state 2 transitions comprise reversible redox-regulated and phosphorylation/dephosphorylation mediated dissociation/association of a mobile fraction of LHCII with the cores of both photosystems, occurring on the order of about 10 min [135]. State transitions tend to be suppressed under high-light conditions in plants [138]. State transitions and other antenna-related regulatory phenomena can be readily measured, even in intact plants and algae, using Chl fluorescence techniques, including pulse-amplitude-modulated (PAM) fluorometric technology [135,138].

A comparatively slow adaptation to varying light conditions is achieved by pigment and/or protein synthesis and degradation taking place on the order of several hours to days.

Excess light energy input often results in a phenomenon referred to as photoinhibition, eventually leading to photodamage and destruction of the photosynthetic apparatus [139]. Excess of excited Chl singlet states, which are not utilized in photochemistry, gives rise to long-lived excited Chl triplet states (^3^Chl*), via intersystem crossing (ISC). ^3^Chl* has the potential to generate free radicals by electron or hydrogen transfer (type I mechanism) or to sensitize the generation of highly reactive singlet oxygen via the interaction with O_2_ in its triplet ground state (type II mechanism) [140]. These photosensitized processes will become harmful when (photo)protective mechanisms cannot sufficiently defuse the reactive pigment and oxygen species.

Photosynthesizing organisms contain carotenoids as essential constituents of LHCs providing effective photoprotection: (i) carotenoids in Van der Waals contact with Chls are very efficient quenchers of ^3^Chl*; in LHCII, the rate of ^3^Chl* quenching by carotenoids is much faster that the formation via ISC, see, e.g., [75,141], and (ii) carotenoids can efficiently defuse singlet oxygen or detoxify other reactive species [12].

In addition to the general protection by the “carotenoid triplet valve” [142], oxygenic photosynthetic organisms have evolved highly efficient mechanisms to alleviate the deleterious effects of excess excitation energy input: the excess population of ^1^Chl* singlets is safely and effectively dissipated nonradiatively as heat by a much studied, but still not fully elucidated, process, which can be assessed as nonphotochemical quenching of Chl fluorescence (NPQ). In plants and algae, the major component of NPQ is the rapidly inducible and relaxing “energy dependent quenching” (qE)—connected to proton translocation into the thylakoid lumen (and resulting “acidification”). The qE mechanism can effectively dissipate excited singlet states of Chl *a* (and, thus, prevent ^3^Chl* formation). The qE mechanism is apparently a complex cooperative phenomenon, which requires several—interdependent—factors. Decisive, however, is the drop of the lumenal pH below ~6, which triggers the following sequence of events: certain amino acid residues in LHCs and in the PsbS protein are protonated, and the respective proteins undergo conformational changes. Violaxanthin (being peripherally bound to LHCII, see Figure 2) is released from its binding pocket. A lumen-localized enzyme, violaxanthin-deepoxidase, is activated by the lowered pH and attaches to the membrane. Violaxanthin is de-epoxidized in two steps (via antheraxanthin, the monoepoxy intermediate) to zeaxanthin [143]. Subsequently, zeaxanthin is assumed to reassociate with LHCs. Upon release of the stress conditions, zeaxanthin is again (via antheraxanthin) enzymatically epoxidized to violaxanthin. Hence, the dark- or low-light reversible process is called the “xanthophyll cycle”.

Numerous studies have shown that qE is intimately correlated with zeaxanthin accumulation [144]. However, the actual *molecular* mechanism of the xanthophyll-dependent dissipation of excess energy remains elusive and is fiercely debated. One model purports that the lowest-lying singlet state (S_1_) of violaxanthin (with nine conjugated double bonds determining the energy of the lowest electron transition in polyenes) is above the S_1_ state of Chl *a*, thus rendering violaxanthin an effective donor of excitation energy to Chl *a* [145]. On the other hand, the transition to the S_1_ state of zeaxanthin (having 11 double bonds) is assumed to be energetically well below that of S_1_ of Chl *a* so that the quenching of this excited state may be feasible. This model is usually referred to as the “molecular gear shift” model [145]. Computationally, it is unclear whether gas phase models suffice for predicting the actual energetic interactions between Chl/xanthophyll chromophore pairs [146], and the problem is also methodologically challenging [147].

Furthermore, a possible quenching mechanism, based on the xanthophyll exchange, was computationally found; however, that mechanism would only remove excess energy from the Chl Soret band before internal conversion to the Q band [148].

Another model proposes that quenching is brought about by structural changes and/or the aggregation of LHCs, with zeaxanthin (in contrast to violaxanthin) giving rise to enhanced aggregation [149]. The considerations of the mechanism of NPQ become even more complicated by the finding that the above mentioned PsbS protein appears also to be critically involved [150]. It has been shown that mutants of *Arabidopsis thaliana* lacking the PsbS protein exhibit a severely impaired capacity for qE [150]. The PsbS protein, although genes of it are found in green algae, is apparently not expressed in Chlamydomonas. PsbS-like function(s) in NPQ in certain green algae were reported to be exerted by LHCSR, another member of the LHC protein family [151]. The possible interaction of PsbS and CP29 was also explored computationally [152]. Combined experimental and computational models propose PsbS to be primarily conveying the luminal acidification signal to a set of amino acids on the stromal side of LHCII. Those amino acids subsequently change their protonation state, giving rise to charge transfer (CT) states in nearby Chls, which in turn allow for radiation-less deactivation of excess singlet excitation. This mechanism would allow LHCII to remain relatively rigid and has the advantage that no specific quenched conformation of the protein structure (which so far eluded all attempts of characterization) would be required [133,134].

An energetic shift of CT states of (strongly) interacting Chls due to a polarity change of the environment induced by structural changes was suggested [153]. CT states seem to play important, not yet fully understood roles in photosynthetic EET processes [154,155].

Another model has been put forward which proposes Chl-to-carotenoid CT with carotenoid radical formation as the mechanistic basis of excess energy dissipation, also involving participation of the PsbS protein [156,157].

Yet, another proposal assumes that altered excitonic interactions between Chls and certain xanthophylls are responsible for the qE mechanism [158]. These experiments were based on the combined study of one- and two-photon excitation in the spectral region of the presumed “dark” S_1_ states of carotenoids/xanthophylls. However, recent two-photon excitation studies of isolated pigments and LHCs indicate predominant two-photon absorption of Chls in the respective spectral region [159]. Hence, a re-evaluation of previously obtained data appears to be required. This endeavor is currently underway in our labs [160,161]. However, data obtained with NLPF comparing trimeric and (slightly) aggregated LHCII seemed to corroborate a change of Chl-xanthophyll interactions to be the origin of NPQ [43].

Notwithstanding all these conflicting mechanistic models, acidification of the thylakoid lumen, operation of the xanthophyll cycle, an intact PSII antenna system and, possibly, the PsbS protein are all well-established factors in short-term plant photoprotection.

Cyanobacteria do not contain plant-type LHCs. In these organisms another type of NPQ has been developed. Excess excitation energy input at the level of phycobilisomes is dissipated in in cyanobacteria in a blue-green light-activated process by the orange carotenoid protein, OCP [61,162], reviewed by Kirilovsky and Kerfeld [163]. OCP is a water soluble ~35 kDa protein, that binds a single ketocarotenoid, e.g., hydroxyechinenone [61]. The blue-green light-induced action of OCP can be monitored as NPQ of phycobilisome fluorescence. The molecular mechanism seems to be a blue-green light-induced conformational change of the OCP-bound carotenoid, reflected in a color change from orange to red (RCP), which is dark reversible. The underlying structural change appears to be a translocation of the carotenoid between the C- and N-terminal domains of the protein and a resultant a change of pigment–protein interactions [164,165]. OCP is thus another example of how carotenoids can be used as structural elements with mechanical function.

In cyanobacteria, full restoration of fluorescence of PBS and, hence, efficient light-harvesting is dependent on the recently discovered fluorescence recovery protein, FRP [166]. 

**Conclusions:** It is clear that light-harvesting antenna complexes not only serve to enlarge the absorption cross sections of the respective photochemical RCs (delivering excitation energy towards them) but are also vitally important in short- and long-term adaptation and regulation of the energy-transforming processes in the photosynthetic apparatus, in response to external and internal conditions. Thus, the structural diversity in photosynthetic antenna “designs” becomes conceivable.

However, an enigma remains: why are so many light-harvesting complexes trimeric (or multiples thereof)? Given the structural, genetic, pigment-binding and functional diversity of LHCs, as presented above, it appears that the trimeric construction principle may be a result of the environment in which these complexes are located in or are attached to, namely, two-dimensional (2D) membrane systems. As discussed above, the fast regulation of light harvesting seems to be crucial to ensure long-term integrity of the photosynthetic apparatus. Hence, the closest possible spatial proximity of the regulatory factors and the RCs is necessary.

The arrangement to form a specific multimer of order N will require a specific angle of 2π/N between the excitation energy donor and acceptor sites of each monomeric complex for optimal EET, which is unlikely to appear randomly. Short chains (such as dimers) thus may be prone to tilt-over. Longer chains may have enough flexibility to form rings of any order of magnitude (such as LH1/LH2; Figure 7 and Figure 8). However, trimers are the smallest imaginable “rings”. The structural polarity of a native membrane obviously does not allow for the complexes to turn “upside down”, in particular if RCs are involved, and retain functionality. Therefore, trimers may simply be the minimal solution to form modular aggregates that allow for (i) stable spatial orientation and (ii) regulatory control of multiple subunits by membrane-related parameters, e.g., ΔpH across the membrane. This may also hold for so-called “soluble” (i.e., non-membrane integral) antenna complexes, such as FMO. FMO is actually tightly appressed to the membrane-intrinsic RCs by the chlorosomes/baseplate proteins in green bacteria.

Another still not fully understood issue is the existence of so-called “red” antenna states (antenna Chl states at longer wavelengths than the respective RCs) which would require up-hill EET to feed the RCs. The issue was recently discussed (but not solved) by a comprehensive review [167].

Based mainly on 2DES spectroscopic results, obtained with a variety of LHCs (as well as RCs), it has been proposed that photosynthetic EET and charge separation may be driven by coherent quantum effects (subsequently dubbed “quantum biology”) [168]. This notion and the interpretation of 2DES data with regard to light harvesting are currently fiercely debated [169] and shall be the topic of another paper.

## 6. Highlights

We review the wide variety of naturally occurring photosynthetic light-harvesting complexes (LHCs) from a structure-function relationship perspective.A common feature of natural light-harnessing systems is a trimeric (or multiples thereof) organization.We provide a simple explanation of this—so far unaccounted for—phenomenon.Natural light-harnessing systems can be used as inspirations for biomimetic assemblies and/or attached to biohybrid devices for sustainable solar energy conversion.

## Figures and Tables

**Figure 1 molecules-26-03378-f001:**
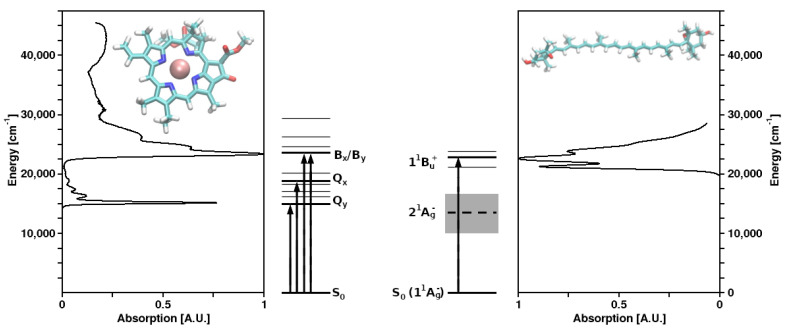
Molecular structures and absorption spectra of chlorophyll *a* (**left**) and the carotenoid (xanthophyll) violaxanthin (**right**) as representatives of their respective molecule classes, as well as the corresponding energy level schemes.

**Figure 2 molecules-26-03378-f002:**
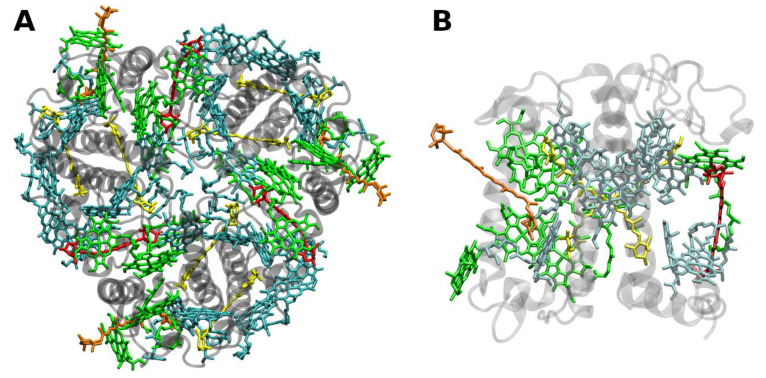
(**A**) Structure of trimeric main light-harvesting complex (LHC II) of plants at 2.5 Å resolution, PDB entry 2BHW [35]. Top view with regard to the thylakoid membrane plane, from the stromal side. (**B**) Monomeric subunit. The protein backbone consisting of three transmembrane α-helical domains is shown in grey. The monomeric subunit noncovalently binds 8 chlorophylls *a* and 6 chlorophylls *b* (shown in cyan and green, respectively) as well as two luteins (yellow), one neoxanthin (orange) and one violaxanthin (red). The binding site of the latter may also be occupied by its de-epoxidation products, antheraxanthin or zeaxanthin.

**Figure 3 molecules-26-03378-f003:**
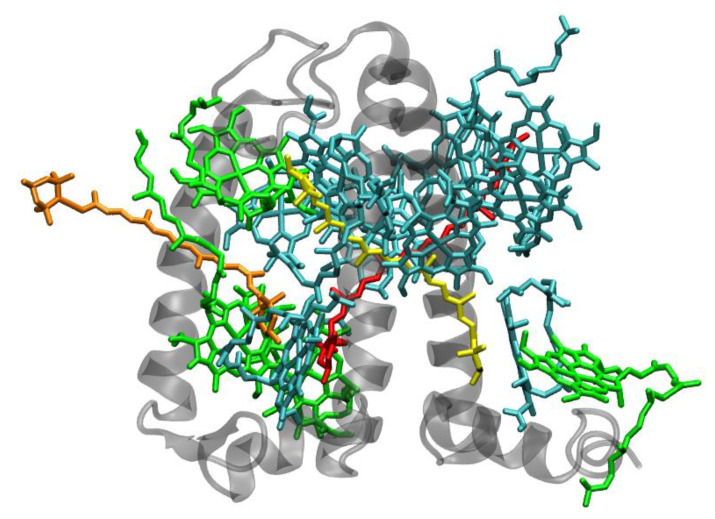
Structure of the monomeric minor light-harvesting complex CP29 of plants at 2.80 Å resolution, PDB entry 3PL9 [36]. Side view with regard to the thylakoid membrane plane. The protein backbone consisting of three transmembrane α-helical domains is shown in grey. CP29 binds 8 chlorophylls *a* and 4 chlorophylls *b* (shown in cyan and green, respectively) and has one possible mixed Chl-binding site. Three xanthophylls are bound, one lutein (yellow), one neoxanthin (orange) and one violaxanthin (red). The binding site of the latter may possibly also be occupied by its de-epoxidation products, antheraxanthin or zeaxanthin.

**Figure 4 molecules-26-03378-f004:**
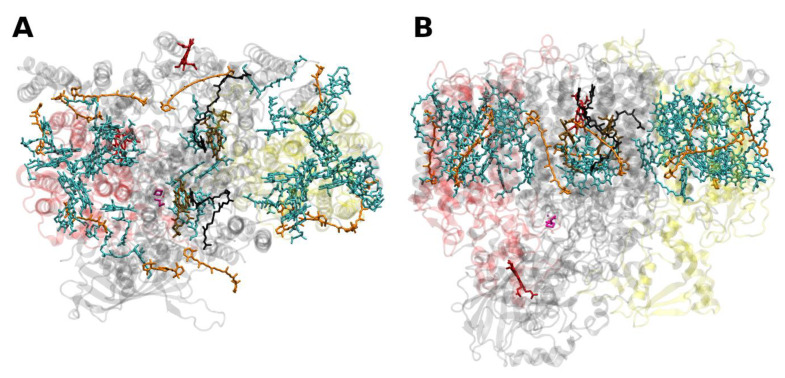
Structure of the photosystem II core with the core-antenna complexes CP43 and CP47 from the thermophilic cyanobacterium *Thermosynecchococcus vulcanus* at 1.9 Å resolution, PDB entry 3ARC [57]. (**A**) Top view with regard to the thylakoid membrane plane, seen from the stromal side. (**B**) Side view with regard to the thylakoid membrane plane. The protein backbone of CP43 (CP47) is shown in red (yellow); other protein subunits are shown in grey. Chlorophylls *a* are shown in cyan, β-carotenes in orange, pheophytin in brown, heme groups in dark red and benzoquinones in grey. The oxygen–manganese cluster of the water splitting apparatus is also shown (violet).

**Figure 5 molecules-26-03378-f005:**
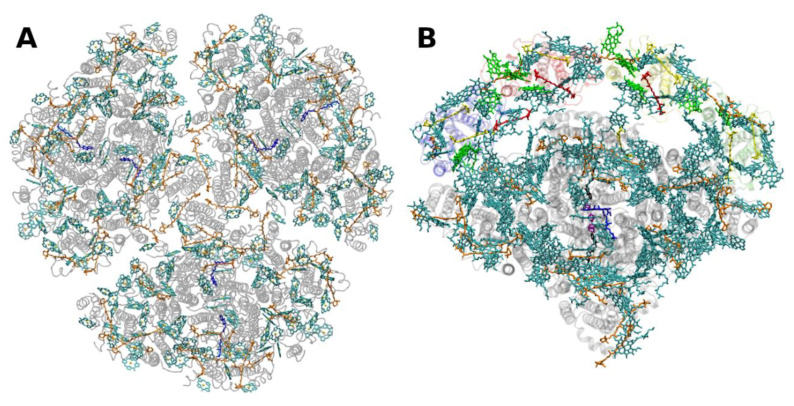
Structure of photosystem I. (**A**): from the thermophilic cyanobacterium *Thermosynecchococcus elongatus* at 2.5 Å resolution, PDB entry 1JB0 [61]. Top view with regard to the thylakoid membrane plane, seen from the stromal side. The protein backbone is shown in grey. Chlorophylls a are shown in cyan, β-carotenes in orange and phylloquinones in grey. (**B**): Structure of monomeric plant photosystem I from *Pisum sativum* with attached LHCI units (Lhca1–4, separated by color) at 2.6 Å resolution, PDB entry 5L8R [32]. Top view with regard to the thylakoid membrane plane, seen from the stromal side. Chlorophylls *a* and *b* are shown in cyan and green, respectively, β-carotenes in orange; the xanthophylls are color-coded as in Figure 1.

**Figure 6 molecules-26-03378-f006:**
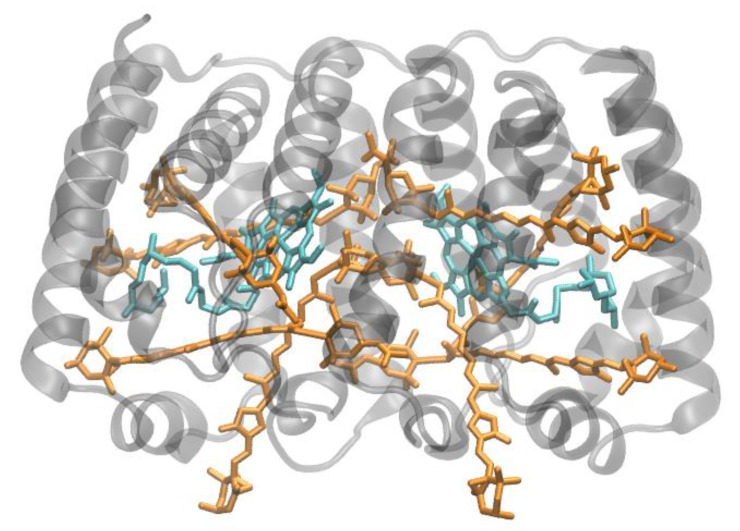
Structure of the trimeric peridin-chlorophyll *a*-protein complex (PCP) from *Amphidinium carterae* at 2.0 Å resolution, PDB entry 1PPR [62]. The protein backbone is shown in grey. Chlorophylls *a* are shown in cyan, peridinins in orange.

**Figure 7 molecules-26-03378-f007:**
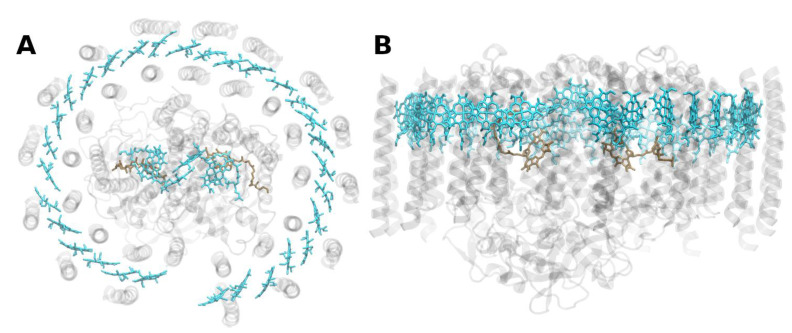
Structure of the purple-bacterial core-antenna reaction center complex (LH1-RC) from *Rhodopseudomonas palustris* at 4.8 Å resolution, PDB entry 1PYH [86]. (**A**) Top view with regard to the bacterial membrane plane, seen from the cytoplasm. (**B**) Side view with regard to the bacterial membrane plane. The protein backbone is shown in grey. Bacteriochlorophyll *a* is shown in light cyan, bacteriopheophytin *a* in brown.

**Figure 8 molecules-26-03378-f008:**
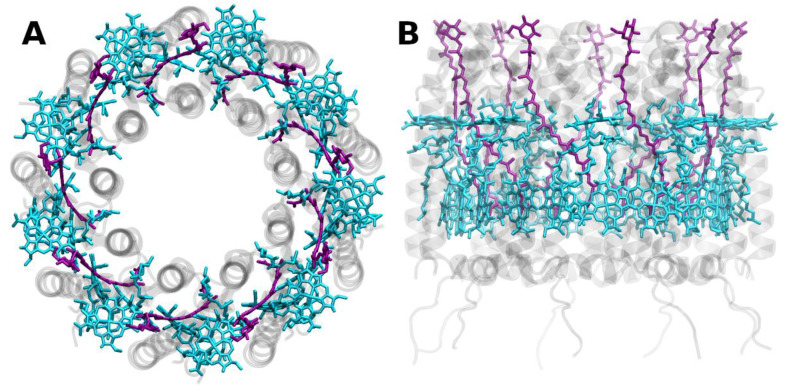
Structure of the purple-bacterial peripheral antenna complex (LH2) from *Rhodoblastus acidophilus* at 2.45 Å resolution, PDB entry 2FKW [87]. (**A**) Top view with regard to the bacterial membrane plane, seen from the cytoplasm. (**B**) Side view with regard to the bacterial membrane plane. The protein backbone is shown in grey. Bacteriochlorophyll *a* is shown in cyan, rhodopin glucosides (carotenoids) in purple.

**Figure 9 molecules-26-03378-f009:**
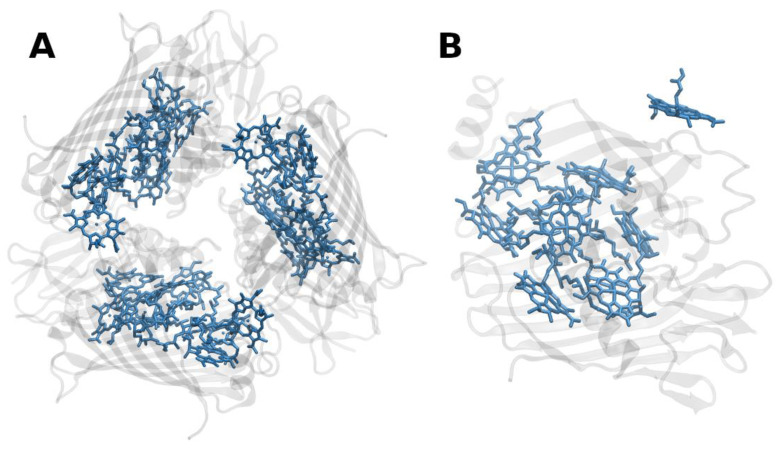
Structure of the water-soluble Fenna–Matthews–Olsen protein complex (FMO) from *Prosthecochloris aestuarii* at 1.3 Å resolution, PDB entry 3EOJ [104]. (**A**) Top view of the trimer with regard to the bacterial membrane plane, seen from the cytoplasm/chlorosome. (**B**) Side view of a single monomer with regard to the bacterial membrane plane. The protein backbone is shown in grey. Bacteriochlorophyll *a* is shown in blue.

**Figure 10 molecules-26-03378-f010:**
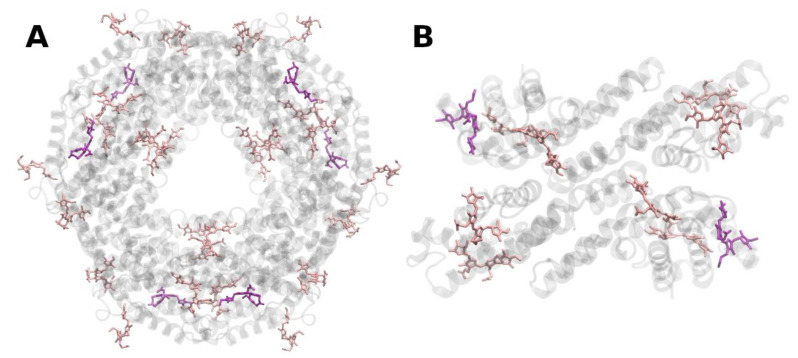
Structure of phycoerythrin from *Gloeobacter violaceus* at 2.19 Å resolution, PDB entry 2VJH [4]. (**A**) View of the dodecamer along the phycobilisome stacking axis. (**B**) Side view of a tetrameric unit, perpendicular to the stacking axis. The protein backbone is shown in grey. Phycoerythrobilins are shown in pink, phycourobilins in violet.

**Figure 11 molecules-26-03378-f011:**
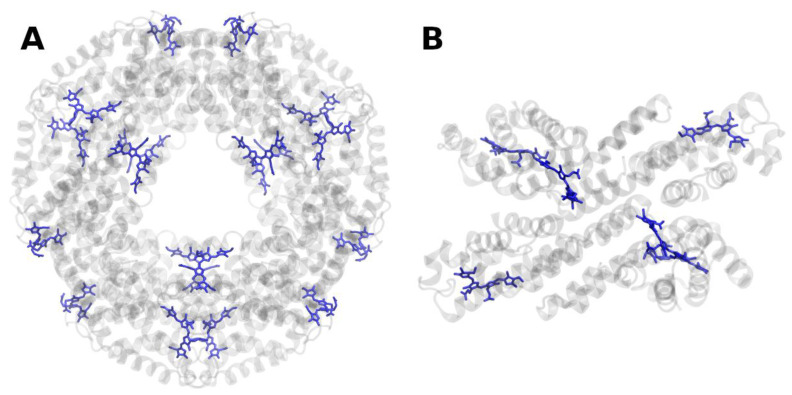
Structure of phycocyanin from *Thermosynechococcus elongatus* at 1.35 Å resolution, PDB entry 3L0F [115]. (**A**) View of the dodecamer along the phycobilisome stacking axis. (**B**) Side view of a tetrameric unit, perpendicular to the stacking axis. The protein backbone is shown in grey. Phycocyanobilins are shown in blue.

## Data Availability

Not applicable.

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
