# Peer review of "Photosynthetic Light-Harvesting (Antenna) Complexes—Structures and Functions"

_molecules, 2021, doi:10.3390/molecules26113378_

Round 1
Reviewer 1 Report
The paper by Lokstein et al. is a comprehensive review about structure and function of light-harvesting complexes. Especially, the authors point out that antenna complexes do not only enlarge the absorption cross-sections of reaction centers, but are also involved in adaptation of the photosynthetic apparatus to external and internal conditions. The review is well written and can be accepted if a number of minor points are appropriately considered by the authors:
1.) on page 4, bottom, it is stated that Chla 612 carries the lowest state of LHCII. However, this is strictly valid at room tempreature only. It has been shown that the position of the absorption band of Chla 612 is temperature dependent (Vrandecic et al. JPCB 2015 https://doi.org/10.1021/jp5112873; Golub et al. JPC B 2018 https://doi.org/10.1021/acs.jpcb.8b02948) and not necessarily identical to the lowest state at 680nm mentioned here, which was detected e.g. by spectral hole burning at 4.2K (see e.g. a recent discussion in: https://doi.org/10.1007/s11120-020-00752-9)
2.) ref. 138 (Kell et al.) is not appropriate to prove CT states by observation of strong electron-phonon coupling. Their work is in error, because the authors of ref. 138 have applied an invalid theory to analyze their data, please check the relevant literature. Ref. 138 should be replaced by a more appropriate reference from spectral hole burning on LHC aggregates. A simple criterion is that in case of CT states/strong coupling no separate ZPL can be observed.
Author Response
We would like to thank all reviewers for their efforts and their constructive remarks on our manuscript, which hopefully have helped to create a better – revised, - version.
R1:
The paper by Lokstein et al. is a comprehensive review about structure and function of light-harvesting complexes. Especially, the authors point out that antenna complexes do not only enlarge the absorption cross-sections of reaction centers, but are also involved in adaptation of the photosynthetic apparatus to external and internal conditions. The review is well written and can be accepted if a number of minor points are appropriately considered by the authors:
1.) on page 4, bottom, it is stated that Chla 612 carries the lowest state of LHCII. However, this is strictly valid at room tempreature only. It has been shown that the position of the absorption band of Chla 612 is temperature dependent (Vrandecic et al. JPCB 2015 https://doi.org/10.1021/jp5112873; Golub et al. JPC B 2018 https://doi.org/10.1021/acs.jpcb.8b02948) and not necessarily identical to the lowest state at 680nm mentioned here, which was detected e.g. by spectral hole burning at 4.2K (see e.g. a recent discussion in: https://doi.org/10.1007/s11120-020-00752-9)
We fully agree with the reviewer, that at low temperatures (77 K) the lowest state of LHCII may not be associated with Chl a 612 anymore. Indeed, our own earlier observations have also suggested this, too, see, e.g. ref. 37. We have also added one ref. suggested by the reviewer.
2.) ref. 138 (Kell et al.) is not appropriate to prove CT states by observation of strong electron-phonon coupling. Their work is in error, because the authors of ref. 138 have applied an invalid theory to analyze their data, please check the relevant literature. Ref. 138 should be replaced by a more appropriate reference from spectral hole burning on LHC aggregates. A simple criterion is that in case of CT states/strong coupling no separate ZPL can be observed.
We do agree and have replaced ref. 138, and added ref. 39 as well.
Reviewer 2 Report
This manuscript by Lokstein et al reviews the current literature on photosynthetic antenna complexes, focusing on structure and function and on computational characterization.
The review is organized by class of organism and antenna complexes, but there is a specific section on photoprotection at the end.
This review attempts at covering the entire spectrum of photosynthetic complexes, encompassing the light-harvesting and photoprotection functions. However, the topic is maybe too broad for a brief review, and many important concepts are left behind. Also the organization of the review seems confused.
I cannot recommend the publication of this review in its current form, but I would be glad to accept a through revised version. I strongly suggest carefully rewriting and re-organizing this review taking into account the points outlined below.
Main issues:
1) This review tries to cover the topic of antenna complexes, and is organized in the following sections: (1) Introduction, (2) photosynthetic apparatus of green plants/algae, (3) antenna complexes of prokaryotes, (4) In-vitro reconstitution of LHCs, (5) Photodamage/Photoprotection. This organization is somewhat confusing to a reader. I understand keeping the discussion of photoprotection in its own section. However, it seems that section 2 is the most elaborate one, whereas the other sections are more rushed. In addition, there is a lot of discussion on the reaction centers, at the beginning of section 2 and again in section 2.2. From the title/abstract/Introduction, this review is focused on the antenna complexes, not on the core complexes or the reaction centers. If the Auhtors want to include also this topic, they should organize the review differently. A few paragraphs on computational modelling are randomly placed in the middle of Section 2.1. Section 4 is very short and contains information only on LHCII/CP29, and as such it could be merged into Section 2. A possibility would be to split the plants/algae section and create a new section on accessory antenna complexes found in algae and cyanobacteria, so that Section 2 would be focused on the Lhc superfamily.
2) A clear description photosynthetic pigments is missing. This missing information generates paradoxical confusion especially when the Authors introduce PCP. There, for the first time, it is said that carotenoids have a dark S1 excited state. However carotenoids are mentioned also earlier for plant antenna complexes. Maybe the Authors could add a section on photosynthetic pigments as done in [Mirkovic et al., Chem Rev 2017 10.1021/acs.chemrev.6b00002], where they briefly explain the excited states of Chls and Cars and their characteristics.
3) Also the concepts of excitonic coupling and EET are taken for granted. Given that these concepts are central to the understanding of antenna complexes, a (brief) explanation appears mandatory. In particular, it should be made clear to the reader what is excitonic coupling, and how it affects spectra (by splitting/shifting energy levels) and dynamics (by inducing EET).
4) The Authors reference mainly earlier works, with limited attention to the advances of the last few years, and with a strong bias towards some techniques, at the expense of others which have surely contributed to the field. On the spectroscopic side, the results gained by the most commonly used techniques (Absorption/CD, TRF, TA, 2DES, single-molecule) have been covered only marginally. On the theoretical/computational side, only few papers have been considered, with little attention to the insight given by computational studies. I understand that it is impossible to comprehensively review all of the advances in photosynthesis, but the Authors should anyway try to give a more balanced review of the literature.
5) As written in the Introduction, the review is focused on "the functional similarities and structural diversity of photosynthetic LHCs" and on "how computational modeling of LHC functions benefits from the availability of high-resolution structures." While the first goal is partially achieved, the second is completely left behind. No further mention on the relation between computational modeling and high-resolution structures is even made. I assume that the Auhtors mainly want to focus on atomistic modeling of antenna complexes, which, as said above, is only marginally discussed. In the end, the reader is left wondering how tightly the computational modeling is linked to the availability of high-resolution structures. Note that there is a growing interest in MD-based studies of antenna complexes, which hold great promise for overcoming the need of high-resolution crystal structures (see 10.1007/s11120-020-00741-y).
6) Introduction, "In plants and algae, LHCs bind exclusively xanthophylls". Do the Authors mean Lhcb proteins? Because Lhcas also bind beta-carotene.
7) Section 2: It would be better to merge the figures of LHCII and CP29, and add a figure describing the organization of PSII. There is a number of recently solved structures of PSII which could be used.
8) Section 2.1 is quite confusing. The Authors present the antenna complexes of PSII and then PSI, then they discuss the sequence similarities among LHCs, but it's not clear if they are talking about Lhcbs only or all Lhc genes. Finally, they go back to LHCII and discuss its pigment composition and organization. They write that " Most Chl a molecules are in intimate contact with xanthophylls providing photoprotection", which is not true. Only four Chl a molecules (a602,a603,a610,a612) are in contact with xanthophylls.
9) The description of computational works in Section 2.1 is unbalanced and acritical. There, it is also stated that "So far, much theoretical research on LHCs has been focused on quantification of excitonic couplings of the chromophores" which is not true. A lot of computational work has been also focused on determining the site energies of the chromophores, and in general on reproducing the spectroscopy of the entire complex (Refs. 44,45,50, 10.1016/j.bbabio.2012.02.016, 10.1021/jp106323e, to mention only Lhcbs). The Authors should also see the reviews focused on theoretical modeling of LHCs (10.1007/s11120-013-9893-3, 10.1016/j.bbabio.2019.07.004). In its current form, this part gives the reader the idea that computational modelling is still very limited.
10) Section 2.3 presents the literature on PCP from a biased point of view. Recent literature agrees that most of Per -> Chl EET occurs mainly through the S2 state, rather than through S1 (10.1021/acs.jpclett.6b02881). These references should be added to the discussion.
11) The section on purple bacteria is also strongly unbalanced. For the spectroscopy of LH2, only a few old papers are referenced. The pump-probe, 2DES, and single-molecule works are missing (e.g. by Engel, van Grondelle, Schlau-Cohen, Köhler, etc.). The Authors report a B850 delocalization length of 16 +/- 4 pigments, which is completely unreasonable, given that the B850 ring has only 18 BChls, and that the degenerate states (the bright ones) can be delocalized at most on 12 pigments. Other estimates of the delocalization length are much smaller, but they are not reported here. Also the computational investigations of LH2 are being ignored (e.g. 10.1021/acs.jpclett.5b00078 10.1039/C9SC02886B). Since, as the Authors state, LH2 is a very nice model system, it could be useful to reference a few purely modellistic papers (10.1021/jp400957t 10.1021/jp202344s 10.1002/cphc.201000913 10.1021/jp108187m).
12) Section 3.3, "Binding constrains the phycobilin [...] and thus, strongly modulates their photophysical properties" This sentence (and the rest of the paragraph) needs some references.
13) Section 4: "Chls, specific xanthophylls [...] are required for the stability of the LHCs in vitro" This part should be expanded: which Chls and which xanthophylls are required for the stability of LHCs?
14) Section 5: The first two paragraphs of this section are messy and confusing, and some concepts are explained better in the following paragraphs. Maybe it would be better to first present the problem of excess light and photodamage, and then discuss the various photoprotection strategies. The Authors are maybe not aware of the computational studies on triplet quenching (10.1021/jp200200x), which should be cited in this context.
15) It is clear that the mechanism of NPQ is very complicated, and still not fully understood. However, it is established that, in high-light conditions, (i) The RCs are "closed" and cannot accept excitation, (ii) as a consequence, long-lived Chl* species remain in the antenna complexes, and (iii) these Chl* can undergo ISC and form triplet Chls. The NPQ site in PSII has been identified with the antenna complexes (either major or minor). In addition, it is also known that isolated antenna complexes can undergo conformational changes and they are found in light-harvesting and quenched conformations (10.1021/acs.jpclett.5b00034 10.1016/j.chempr.2019.08.002 10.1104/pp.15.01935 10.1038/srep15661). In vivo, the quenched conformation is induced by PsbS, which is the protein that senses the lumenal pH. All of this occurs also without the need for zeaxanthin.
16) The exact quenching mechanism, and what species actually quenches Chl excitation, is another, separate (but strongly related to the above) issue, which is far from solved. However, the most commonly accepted quenching mechanism is EET from the Chl to the S1 state of xanthophylls (i.e. Lut or Zea), which is supported by TA and 2DES measurements on both isolated complexes and thylakoid membranes (see e.g. 10.1038/s41467-020-15074-6 10.1016/j.chempr.2019.08.002). Understandably, this mechanism was also the most thoroughly studied computationally (By Duffy and co-workers mainly, see e.g. 10.1039/c5cp01905b). The molecular gear-shift mechanism assumes the EET quenching mechanism to be the most important one. The alternative CT mechanism has also been studied computationally .
17) The hypothesis of excitonic state between Qy and S1, put forward on the basis of certain TA signals, clashes with the nature of the S1 state. The exciton coupling between Qy and S1 is very small (at most few tens of cm^-1), and the reorganization energy of the carotenoids is very large (thousands of cm^-1). On this basis, exciton delocalization between Qy and S1 would be impossible.
18) At the end of section 5, it is stated that for OCP "The exact molecular mechanism is currently not known, but blue-green light induces a conformational change of the OCP-bound carotenoid reflected in a colour change from orange to red, which is dark-reversible". This is not completely correct. Recent studies have shown that the carotenoid translocates into the NTD of OCP after breaking the H-bonds with the CTD. RCP, i.e. the NTD of OCP binding the carotenoid, is considered as a model of the red form of OCP.
Minor points:
a) There are many parentheses, in the title, abstract, and Introduction, which make reading these parts quite cumbersome. Consider expanding or rewriting some expressions.
b) Cryptophyte algae also contain phycobilisomes.
Indeed, there are many experimental and computational investigations on phycocyanins from cryptophyte algae, maybe more than on cyanobacterial ones. The Auhtors might consider adding also these systems to the review.
c) Page 8 "A significant blue-shift [...] can obviously not be explained by the [...] non-equivalence of the Chls a." There is nothing obvious here. Since the NLPF technique is obscure to most readers, it is not clear what the results imply. The Auhtors should rephrase this, e.g. as "NLPF experiments have suggested that [...]".
d) "PG is assumed in binding a Chl a [...]" This sentence is unclear and should be rephrased
e) Page 13, "This model was challenged by computational approaches [...]". The computational approaches used in ref. 130 were not appropriate for describing the S1 state of Cars (which is a doubly excited state).
f) Page 14, "CT states seem to play important [...] roles in photosynthetic EET processes". The Author should look at the following references: 10.1039/C7CP03038J 10.1039/c3cp54634a 10.1021/acs.jpclett.8b03233
g) Page 15, "why do so many light-harvesting complexes are trimeric (or multiples thereof)? " This sentence is grammatically incorrect, it should be "why are so many light-harvesting complexes trimeric (or multimeric)?". Note that some LH2 complexes are 7-meric or 8-meric.
h) "The functions and photophysics of carotenoids in LHCs have been covered extensively in a previous review in this journal [12]". Ref [12] is not in Molecules, but in J. Photochem. Photobiol. C Photochem. Rev.
Author Response
R2:
We would like to thank all reviewers, in particular this one, for their extensive efforts and their constructive remarks on our manuscript, which hopefully have helped to create a better – revised, - version.
This manuscript by Lokstein et al reviews the current literature on photosynthetic antenna complexes, focusing on structure and function and on computational characterization. The review is organized by class of organism and antenna complexes, but there is a specific section on photoprotection at the end.
This review attempts at covering the entire spectrum of photosynthetic complexes, encompassing the light-harvesting and photoprotection functions. However, the topic is maybe too broad for a brief review, and many important concepts are left behind. Also the organization of the review seems confused.
I cannot recommend the publication of this review in its current form, but I would be glad to accept a through revised version. I strongly suggest carefully rewriting and re-organizing this review taking into account the points outlined below.
We agree with the reviewer, that not all aspects of light-harvesting and photoprotection were covered fully, however, focusing mainly on structural aspects made this manuscript already quite lengthy. It was not our intention to provide a review on the molecular control and excitation energy transfer mechanisms.
Main issues:
1) This review tries to cover the topic of antenna complexes, and is organized in the following sections: (1) Introduction, (2) photosynthetic apparatus of green plants/algae, (3) antenna complexes of prokaryotes, (4) In-vitro reconstitution of LHCs, (5) Photodamage/Photoprotection. This organization is somewhat confusing to a reader. I understand keeping the discussion of photoprotection in its own section. However, it seems that section 2 is the most elaborate one, whereas the other sections are more rushed. In addition, there is a lot of discussion on the reaction centers, at the beginning of section 2 and again in section 2.2. From the title/abstract/Introduction, this review is focused on the antenna complexes, not on the core complexes or the reaction centers. If the Auhtors want to include also this topic, they should organize the review differently. A few paragraphs on computational modelling are randomly placed in the middle of Section 2.1. Section 4 is very short and contains information only on LHCII/CP29, and as such it could be merged into Section 2. A possibility would be to split the plants/algae section and create a new section on accessory antenna complexes found in algae and cyanobacteria, so that Section 2 would be focused on the Lhc superfamily.
We do not agree with the reviewer that the organization of the article should be changed entirely. A short mentioning of reaction centers (RCs) had to be included to explain the function of LHCs, although the RCs are not the core topic of the review. However, the manuscript order was somewhat rearranged (see paras marked in yellow) to follow the reviewer’s suggestions.
2) A clear description photosynthetic pigments is missing. This missing information generates paradoxical confusion especially when the Authors introduce PCP. There, for the first time, it is said that carotenoids have a dark S1 excited state. However carotenoids are mentioned also earlier for plant antenna complexes. Maybe the Authors could add a section on photosynthetic pigments as done in [Mirkovic et al., Chem Rev 2017 10.1021/acs.chemrev.6b00002], where they briefly explain the excited states of Chls and Cars and their characteristics.
In principle, we agree with the reviewer, however, for the sake of brevity, we decided to omit extensive coverage, since this is a Special Issue on Chls and carotenoids, which are/will be being dealt with in other articles rather extensively and this manuscript is already quite lengthy. However, we have added a new Fig. 1 covering basic structural and spectroscopic properties of Chls and carotenoids.
3) Also the concepts of excitonic coupling and EET are taken for granted. Given that these concepts are central to the understanding of antenna complexes, a (brief) explanation appears mandatory. In particular, it should be made clear to the reader what is excitonic coupling, and how it affects spectra (by splitting/shifting energy levels) and dynamics (by inducing EET).
We agree with the reviewer, and have added a respective paragraph in the revised MS.
4) The Authors reference mainly earlier works, with limited attention to the advances of the last few years, and with a strong bias towards some techniques, at the expense of others which have surely contributed to the field. On the spectroscopic side, the results gained by the most commonly used techniques (Absorption/CD, TRF, TA, 2DES, single-molecule) have been covered only marginally. On the theoretical/computational side, only few papers have been considered, with little attention to the insight given by computational studies. I understand that it is impossible to comprehensively review all of the advances in photosynthesis, but the Authors should anyway try to give a more balanced review of the literature.
We have avoided extensive discussion of more recent spectroscopic work on purpose, in particular with regard to two-dimensional electronic spectroscopy (2DES). Moreover, interpretation of oscillations in 2DES data is highly contentious, to say at least, and a proper discussion of this issue is way beyond this review. There are several extensive recent opinionated reviews of the issue available. Moreover, a manuscript detailing the views of one of the authors on this issue is currently in preparation and will be published elsewhere.
5) As written in the Introduction, the review is focused on "the functional similarities and structural diversity of photosynthetic LHCs" and on "how computational modeling of LHC functions benefits from the availability of high-resolution structures." While the first goal is partially achieved, the second is completely left behind. No further mention on the relation between computational modeling and high-resolution structures is even made. I assume that the Auhtors mainly want to focus on atomistic modeling of antenna complexes, which, as said above, is only marginally discussed. In the end, the reader is left wondering how tightly the computational modeling is linked to the availability of high-resolution structures. Note that there is a growing interest in MD-based studies of antenna complexes, which hold great promise for overcoming the need of high-resolution crystal structures (see 10.1007/s11120-020-00741-y).
We thank the reviewer for pointing out this neglect. We have added a discussion on the modeling topic throughout the new version of the article. We have also re-arranged several paragraphs to place theory more at the end of each section. The cited reference is a good review article, we have, however, trouble sharing the optimistic view on the current potential of MD-derived structure(s). Two examples to make our point clearer: (i) Carotenoid structures and classic MD do not mix well, since any change of the environment of the carotenoid(s) will not be properly translated to the conjugated (double-)bond structure without a full electronic description. (ii) The triphasic interaction at the membrane surface (protein/solvent/membrane phase) is still not solved on an atomistic level; classic MD results in single water molecules migrating between the phases (see, e.g., the recent work by Cui, http://sites.bu.edu/cui-group/papers/ for example).
6) Introduction, "In plants and algae, LHCs bind exclusively xanthophylls". Do the Authors mean Lhcb proteins? Because Lhcas also bind beta-carotene.
Indeed, we mainly mean Lhcb proteins. Moreover, we think that occurrence of β-carotene in plant LHCI might be an isolation artifact.
7) Section 2: It would be better to merge the figures of LHCII and CP29, and add a figure describing the organization of PSII. There is a number of recently solved structures of PSII which could be used.
In this review we would like to focus on LHCs. Thus, we would like to avoid inclusion of structures of entire photosystems (except for PSI, which is a RC with a fused not to be separated core-antenna system).
8) Section 2.1 is quite confusing. The Authors present the antenna complexes of PSII and then PSI, then they discuss the sequence similarities among LHCs, but it's not clear if they are talking about Lhcbs only or all Lhc genes. Finally, they go back to LHCII and discuss its pigment composition and organization. They write that " Most Chl a molecules are in intimate contact with xanthophylls providing photoprotection", which is not true. Only four Chl a molecules (a602,a603,a610,a612) are in contact with xanthophylls.
We would like to thank the reviewer, this has been corrected in the revised MS.
9) The description of computational works in Section 2.1 is unbalanced and acritical. There, it is also stated that "So far, much theoretical research on LHCs has been focused on quantification of excitonic couplings of the chromophores" which is not true. A lot of computational work has been also focused on determining the site energies of the chromophores, and in general on reproducing the spectroscopy of the entire complex (Refs. 44,45,50, 10.1016/j.bbabio.2012.02.016, 10.1021/jp106323e, to mention only Lhcbs). The Authors should also see the reviews focused on theoretical modeling of LHCs (10.1007/s11120-013-9893-3, 10.1016/j.bbabio.2019.07.004). In its current form, this part gives the reader the idea that computational modelling is still very limited.
This part was amended according to the reviewers’ suggestions.
10) Section 2.3 presents the literature on PCP from a biased point of view. Recent literature agrees that most of Per -> Chl EET occurs mainly through the S2 state, rather than through S1 (10.1021/acs.jpclett.6b02881). These references should be added to the discussion.
In principle, we do agree with the reviewer. However, also in the quoted ref. 58 EET from Per S2 to Chl a is envisioned as being dominant.
11) The section on purple bacteria is also strongly unbalanced. For the spectroscopy of LH2, only a few old papers are referenced. The pump-probe, 2DES, and single-molecule works are missing (e.g. by Engel, van Grondelle, Schlau-Cohen, Köhler, etc.). The Authors report a B850 delocalization length of 16 +/- 4 pigments, which is completely unreasonable, given that the B850 ring has only 18 BChls, and that the degenerate states (the bright ones) can be delocalized at most on 12 pigments. Other estimates of the delocalization length are much smaller, but they are not reported here. Also the computational investigations of LH2 are being ignored (e.g. 10.1021/acs.jpclett.5b00078 10.1039/C9SC02886B). Since, as the Authors state, LH2 is a very nice model system, it could be useful to reference a few purely modellistic papers (10.1021/jp400957t 10.1021/jp202344s 10.1002/cphc.201000913 10.1021/jp108187m).
We do not agree with the reviewer, that the section is strongly unbalanced. Moreover, we have avoided an extensive coverage of purple bacterial LH complexes, since these model systems are covered extensively in many recent reviews. Moreover, we do not think that a B850 delocalization length of 16 +/- 4 pigments is completely unreasonable, because it was actually measured. Some other estimates do not seem to be reliable (at least to us), e.g., using super-radiance: These will indicate values after at least some exciton relaxation.
12) Section 3.3, "Binding constrains the phycobilin [...] and thus, strongly modulates their photophysical properties" This sentence (and the rest of the paragraph) needs some references.
This is meanwhile well established textbook knowledge.
13) Section 4: "Chls, specific xanthophylls [...] are required for the stability of the LHCs in vitro" This part should be expanded: which Chls and which xanthophylls are required for the stability of LHCs?
Lutein is required to form stable trimers (as noted in the manuscript), Chl b is needed for stable LHCII assembly in general.
14) Section 5: The first two paragraphs of this section are messy and confusing, and some concepts are explained better in the following paragraphs. Maybe it would be better to first present the problem of excess light and photodamage, and then discuss the various photoprotection strategies. The Authors are maybe not aware of the computational studies on triplet quenching (10.1021/jp200200x), which should be cited in this context.
In fact, one of the authors has done experimental not computational studies; even temperature dependent ones, on triplet quenching with these complexes, see ref. [70]. A theory reference was added to support the statement of a“triplet valve”.
15) It is clear that the mechanism of NPQ is very complicated, and still not fully understood. However, it is established that, in high-light conditions, (i) The RCs are "closed" and cannot accept excitation, (ii) as a consequence, long-lived Chl* species remain in the antenna complexes, and (iii) these Chl* can undergo ISC and form triplet Chls. The NPQ site in PSII has been identified with the antenna complexes (either major or minor). In addition, it is also known that isolated antenna complexes can undergo conformational changes and they are found in light-harvesting and quenched conformations (10.1021/acs.jpclett.5b00034 10.1016/j.chempr.2019.08.002 10.1104/pp.15.01935 10.1038/srep15661). In vivo, the quenched conformation is induced by PsbS, which is the protein that senses the lumenal pH. All of this occurs also without the need for zeaxanthin.
We are fully aware of all these facts, however, this is not a review on NPQ. We try to show that certain LHCs do not only deliver excitation energy to the RCs but are also involved in photoprotection/regulation of EET to the RCs.
16) The exact quenching mechanism, and what species actually quenches Chl excitation, is another, separate (but strongly related to the above) issue, which is far from solved. However, the most commonly accepted quenching mechanism is EET from the Chl to the S1 state of xanthophylls (i.e. Lut or Zea), which is supported by TA and 2DES measurements on both isolated complexes and thylakoid membranes (see e.g. 10.1038/s41467-020-15074-6 10.1016/j.chempr.2019.08.002). Understandably, this mechanism was also the most thoroughly studied computationally (By Duffy and co-workers mainly, see e.g. 10.1039/c5cp01905b). The molecular gear-shift mechanism assumes the EET quenching mechanism to be the most important one. The alternative CT mechanism has also been studied computationally .
We are fully aware of all these facts (see above), however, this is not a review on NPQ, and the suggested references would shift the focus of our review towards NPQ. We are, of course, ourselves interested in NPQ, but we specifically do not want it to be the core topic in this review.
17) The hypothesis of excitonic state between Qy and S1, put forward on the basis of certain TA signals, clashes with the nature of the S1 state. The exciton coupling between Qy and S1 is very small (at most few tens of cm^-1), and the reorganization energy of the carotenoids is very large (thousands of cm^-1). On this basis, exciton delocalization between Qy and S1 would be impossible.
We do, indeed, not claim that excitonic interaction(s) between Chl Qy and carotenoid S1 are the (major) basis of NPQ (as do, e.g., Bode et al.). However, we have previously experimentally observed a change of interaction(s), not necessarily excitonic ones, between certain Chls a and b and carotenoid(s) upon aggregation of LHCII (supposed to be related to the induction of quenching). See also refs. 40/41.
18) At the end of section 5, it is stated that for OCP "The exact molecular mechanism is currently not known, but blue-green light induces a conformational change of the OCP-bound carotenoid reflected in a colour change from orange to red, which is dark-reversible". This is not completely correct. Recent studies have shown that the carotenoid translocates into the NTD of OCP after breaking the H-bonds with the CTD. RCP, i.e. the NTD of OCP binding the carotenoid, is considered as a model of the red form of OCP.
Whilst we appreciate such progresses in elucidation of the mechanism(s), it is our opinion that other explanations may be still possible. Hence, we consider the mechanism as still under dispute.
Minor points:
- a) There are many parentheses, in the title, abstract, and Introduction, which make reading these parts quite cumbersome. Consider expanding or rewriting some expressions.
We have rephrased some sentences in the revised MS.
- b) Cryptophyte algae also contain phycobilisomes. Indeed, there are many experimental and computational investigations on phycocyanins from cryptophyte algae, maybe more than on cyanobacterial ones. The Auhtors might consider adding also these systems to the review.
Cryptophyte algae are rather obscure (and ecologically far less important) organisms, their phycobiliproteins are rather similar to the way more important cyanobacteria.
- c) Page 8 "A significant blue-shift [...] can obviously not be explained by the [...] non-equivalence of the Chls a." There is nothing obvious here. Since the NLPF technique is obscure to most readers, it is not clear what the results imply. The Auhtors should rephrase this, e.g. as "NLPF experiments have suggested that [...]".
Readers should also refer to the original papers, which obviously explain these observations - otherwise they would not have been accepted for publication. However, we have rephrased the original sentences accordingly.
- d) "PG is assumed in binding a Chl a [...]" This sentence is unclear and should be rephrased
This statement was taken from the original publication and seems to need no further explanation.
- e) Page 13, "This model was challenged by computational approaches [...]". The computational approaches used in ref. 130 were not appropriate for describing the S1 state of Cars (which is a doubly excited state).
While the authors in the respective paper did not use an appropriate description of the S1 state, their point still stands: Theoretical approaches have so far not found a proper coupling between Qy and S1. Ref 141 (in the new version of the paper) shows that S1 is never energetically below Qy unless it is in a S2 relaxed geometry, at which point it is unavailable to accept excitations from Qy, due to being in the S2 state. Hence an alternative was proposed to explain the quenched signal. This paper uses the state-of-the-art DFT/MRCI method.
- f) Page 14, "CT states seem to play important [...] roles in photosynthetic EET processes". The Author should look at the following references: 10.1039/C7CP03038J 10.1039/c3cp54634a 10.1021/acs.jpclett.8b03233
We appreciate the suggestions, but for the sake of brevity, we have restricted our paper to the cited references.
- g) Page 15, "why do so many light-harvesting complexes are trimeric (or multiples thereof)? " This sentence is grammatically incorrect, it should be "why are so many light-harvesting complexes trimeric (or multimeric)?". Note that some LH2 complexes are 7-meric or 8-meric.
This part was amended accordingly for hopefully better clarity in the revised MS.
- h) "The functions and photophysics of carotenoids in LHCs have been covered extensively in a previous review in this journal [12]". Ref [12] is not in Molecules, but in J. Photochem. Photobiol. C Photochem. Rev.
We would like to thank the reviewer, once again, this has been corrected in the revised MS.
Reviewer 3 Report
This is a useful, timely, quite comprehensive review on a constantly expanding, inherently very broad field of research, the photosynthetic light harvesting complexes (LHCs).
In general, the paper is well written and provides a good overview on the structure and functions of LHCs in virtually all photosynthetic organisms, from different types of anoxygenic bacterial organisms to a wide range of oxygenic organisms, cyanobacteria, different classes of algae and green plants.
With this said, inevitably, several areas are covered better and others (luckily only a few) contain less up-to-date (or seldom misleading) information. Listed below, some points which should and/or might be improved during the revision of the manuscript.
1/ In Section 2, the authors introduce the basic features of plant chloroplast ultrastructure and dynamics. However, regarding the 3D organization of the granum-stroma thylakoid membrane assembly and its structural dynamics key informations are missing. Cf. e.g Bussi et al. 2019 PNAS, Lambrev and Akhtar 2019 Biochem J - and perhaps some references in these papers.. (Reading only refs 13 and 14 would mislead the reader.)
2/ I think that 2D (and 3D) spectroscopy, especially on LHCII would require more attention - taking into account their potential to uncover the pathways of the excitation energy; also mentioning the novel spectroscopic tool of ACD (Anisotropic CD) (Nielsen et al. 2016 Nat Comm; Akhtar et al. 2019 J Phys Chem B), as a selective tool to identify the origin of excitonic interactions, would probably interest the readers of Molecules.
3/ While the authors deal in depth with the molecular architecture of LHCII and roles of LHCII in different regulatory processes, somewhat surprisingly they do not take into consideration its inherent structural flexibility - either influenced by exposing the complexes to different physico-chemical environments, or their ability to readily undergo light-induced reorganizations.
4/ Another field where I feel that the reference list (and content) should be updated is the state transitions.
5/ In the Summary the authors argue that "trimers may simply be the minimal solution to form modular aggregates that allow for (i) stable spatial orientation and, (ii) regulatory control of multiple subunits by membrane-related parameters, e.g. ΔpH across the membrane." I am not convinced about the validity of these statements: (i) known examples of trimerization of water soluble protein complexes - e.g. FMO; (ii) I am not at all convinced that trimerization is needed for the proper insertion of complexes in the membrane or yet it would aid their stable spatial orientation; (iii) PSI is found, in different organisms, both in monomeric and trimeric froms - with virtually identical functions; (iv) instead of trimers, dimers are also often observed - e.g. PSII; (v) I am not sure what the authors mean by referring to the role of trimers in relation to ΔpH.
6/ For the case LHCII trimers, it is interesting to note the physiologically important fact that "the LHCII apoprotein and the monomeric form of the holoprotein are targeted for proteolysis while the trimeric form is not" (Yang et al. 2000 FEBS Lett).
7/ Minor comments and corrigenda:
- Last sentence of the Abstract - unclear if it is for the trimers (or sg else), rephrase or reconsider (see 5/)
- l. 54: certain Cyanobacteria to growth -- certain cyanobacteria to grow
- ll. 231/232: of Chls awhich may constitute as the so called -- of Chls a, which may constitute the so-called
- l. 235: thermophilic Cyanobacterium -- thermophilic cyanobacterium
- l. 238: , phyllochinones -- , phylloquinones
- l. 376: to be (2x)
- l. 418: Cyanobacteria -- cyanobacteria
- l. 560: in (2x)
Author Response
R3:
We would like to thank all reviewers for their efforts and their constructive remarks on our manuscript, which hopefully have helped to create a better - revised - version.
This is a useful, timely, quite comprehensive review on a constantly expanding, inherently very broad field of research, the photosynthetic light harvesting complexes (LHCs).
In general, the paper is well written and provides a good overview on the structure and functions of LHCs in virtually all photosynthetic organisms, from different types of anoxygenic bacterial organisms to a wide range of oxygenic organisms, cyanobacteria, different classes of algae and green plants.
With this said, inevitably, several areas are covered better and others (luckily only a few) contain less up-to-date (or seldom misleading) information. Listed below, some points which should and/or might be improved during the revision of the manuscript.
1/ In Section 2, the authors introduce the basic features of plant chloroplast ultrastructure and dynamics. However, regarding the 3D organization of the granum-stroma thylakoid membrane assembly and its structural dynamics key informations are missing. Cf. e.g Bussi et al. 2019 PNAS, Lambrev and Akhtar 2019 Biochem J - and perhaps some references in these papers.. (Reading only refs 13 and 14 would mislead the reader.)
This part was amended in the revised MS.
2/ I think that 2D (and 3D) spectroscopy, especially on LHCII would require more attention - taking into account their potential to uncover the pathways of the excitation energy; also mentioning the novel spectroscopic tool of ACD (Anisotropic CD) (Nielsen et al. 2016 Nat Comm; Akhtar et al. 2019 J Phys Chem B), as a selective tool to identify the origin of excitonic interactions, would probably interest the readers of Molecules.
We have avoided extensive discussion of two-dimensional electronic spectroscopy (2DES) on purpose (see also above). Moreover, interpretation of oscillations in 2DES data is highly contentious, to say at least, and a proper discussion of this issue is way beyond this review. A manuscript detailing the views of one of the authors on this issue is currently in preparation and will be published elsewhere.
3/ While the authors deal in depth with the molecular architecture of LHCII and roles of LHCII in different regulatory processes, somewhat surprisingly they do not take into consideration its inherent structural flexibility - either influenced by exposing the complexes to different physico-chemical environments, or their ability to readily undergo light-induced reorganizations.
We are fully aware of all these data. We have even mentioned some of our earlier related work (see refs.). However, this is not a review on LHCII alone, and some of these recent studies appear to be highly contentious.
4/ Another field where I feel that the reference list (and content) should be updated is the state transitions.
We don’t think there was much (relevant) news regarding state transitions (in plants) recently …
5/ In the Summary the authors argue that "trimers may simply be the minimal solution to form modular aggregates that allow for (i) stable spatial orientation and, (ii) regulatory control of multiple subunits by membrane-related parameters, e.g. ΔpH across the membrane." I am not convinced about the validity of these statements: (i) known examples of trimerization of water soluble protein complexes - e.g. FMO; (ii) I am not at all convinced that trimerization is needed for the proper insertion of complexes in the membrane or yet it would aid their stable spatial orientation; (iii) PSI is found, in different organisms, both in monomeric and trimeric froms - with virtually identical functions; (iv) instead of trimers, dimers are also often observed - e.g. PSII; (v) I am not sure what the authors mean by referring to the role of trimers in relation to ΔpH.
We would like to thank the reviewer. This part was amended according to the reviewer’s suggestions.
6/ For the case LHCII trimers, it is interesting to note the physiologically important fact that "the LHCII apoprotein and the monomeric form of the holoprotein are targeted for proteolysis while the trimeric form is not" (Yang et al. 2000 FEBS Lett).
We have considered adding the notion, but given that we want to keep the section short, we decided against it, since it seems not to be too relevant here.
7/ Minor comments and corrigenda:
- Last sentence of the Abstract - unclear if it is for the trimers (or sg else), rephrase or reconsider (see 5/)
- l. 54: certain Cyanobacteria to growth -- certain cyanobacteria to grow
- ll. 231/232: of Chls awhich may constitute as the so called -- of Chls a, which may constitute the so-called
- l. 235: thermophilic Cyanobacterium -- thermophilic cyanobacterium
- l. 238: , phyllochinones -- , phylloquinones
- l. 376: to be (2x)
- l. 418: Cyanobacteria -- cyanobacteria
- l. 560: in (2x)
We would like to thank the reviewer, the errors are corrected in the revised MS.
Reviewer 4 Report
This is a comprehensive review article on naturally evolved light harvesting antennae proteins from a structure function relationship perspective. This will be good reference for both young and experienced scientists.
Author Response
R4:
This is a comprehensive review article on naturally evolved light harvesting antennae proteins from a structure function relationship perspective. This will be good reference for both young and experienced scientists.
We would like to thank the reviewer for endorsing our manuscript.
Round 2
Reviewer 2 Report
The Authors improved their review by addressing several of my points. They changed the title in order to remove the reference to computational studies. They generally improved the organization and readability of the review. However, I still think that the review is somewhat unbalanced. I do understand that it is not possible to cover all aspects of light-harvesting complexes in a short review. However, the Authors should strive to give a more balanced view of the literature (I) in presenting the experimental techniques, and (ii) when there is some debate related to the characteristics or function of the LHCs. For this reason, I believe that the manuscript should be further revised.
-
I understand that the Authors do not want to present all the experimental techniques used to understand the structure/function of LHCs. And I agree that the interpretation of 2DES is complex. However, the Authors should at least refer to the most recent review articles on the spectroscopic methods.
-
Similarly, the Authors should acknowledge that MD simulations are now a fundamental tool to understand the structure and dynamics of LHCs, or to explore complex-complex interactions, even though they obviously are not perfect. The reasoning on the limitations of MD simulations is anyway specious, because (a) not all LHCs are membrane complex or contain carotenoids, (b) there exist ad-hoc force fields for carotenoid molecules, as well as Chls, (c) MD simulations can serve as a way to explore the conformational dynamics of LHCs, even if the force fields for the chromophores are not perfect. Also I do not understand the reference to the work by Cui and co-workers: the group actually uses MD simulations to understand the role of water in the conformational dynamics of membrane proteins.
-
I do not agree with the Authors on the delocalization length of the B850 ring. They deem all experimental results as unreliable, except for the one they chose. The latter one was determined from the enhancement of the transition dipole moment and comparison with very simple (exciton-only) modelling. Such combination of experiment and theory cannot distinguish a (pure) exciton delocalized state from a statistical mixing of states, and moreover it cannot separate the k=1 and k=-1 states (only one S0→S1 band is seen in the experiments). Therefore, these experiments only see the combined oscillator strength from more than one excited state. Nonetheless, experiments with several other techniques (in absorption, not in fluorescence) have measured a much shorter delocalization length [10.1021/jp9844415 10.1021/jp953639b ] and show that static disorder mixes the states of the perfect ring [ 10.1016/j.bpj.2014.03.023 ].
-
Another issue is on the OCP mechanism. The Authors are ignoring all the recent studies that support the translocation mechanism [10.1126/science.aaa7234 10.1016/j.tplants.2019.09.013 10.1073/pnas.1512240112 10.1021/jacs.7b05193 10.1038/s41598-017-15520-4 10.1021/jacs.8b11373 10.1038/s41598-020-68463-8 ], thus giving a biased view of the literature on OCP. As of today, the literature agrees on the translocation mechanism. Note that there exist CTD/NTD homologous that are able to exchange bound carotenoids, which serves as an additional indirect proof of the translocation.
-
If the Authors have a reference on the fact that “β-carotene in plant LHCI might be an isolation artifact”, they should cite it; otherwise, they should correct the sentence “In plants and algae, LHCs bind exclusively xanthophylls”
-
I understand that this is not a review on NPQ, but the Authors should try to be more objective in their writing. Firstly, as I already pointed out, ref [139] does not use the correct QM tools to describe the S1 state of carotenoids. In addition, the fact that “S1 is never energetically below Qy unless it is in a S2 relaxed geometry [...]” is completely irrelevant. What is important is that the adiabatic S1 energy is lower than the Qy one, in which case the driving force for EET is negative, and the energy transfer is thermodynamically favoured by virtue of detailed balance considerations. Computational methods aside, the Authors should acknowledge that the spectroscopic literature (see my previous report) does support the population of a Car S1 state during quenching.
-
In my opinion, it is important to mention that the isolated antenna complexes can switch between several conformations. There is independent evidence of this from single-molecule spectroscopy and transient absorption. This fact implies that the crystallographic view of LHCs is probably limited to a single structure in an ensemble of conformations.
-
The sentence “However, data obtained with NLPF on trimeric and aggregated LHCII seem to corroborate a Chl-xanthophyll excitonic interaction origin of NPQ [41].” seems to imply that there is an excitonic mixing of Qy and S1. Based on the Authors’ resonse to my previous report, it should be corrected by noting that the NLPF experiments show a change of interaction between Chls and Cars, but not necessarily an exciton state.
-
The sentence “PG is assumed in binding a Chl a which is supposed to be involved in EET to CP29 […]” should be rephrased because it is not clear. The fact that it was taken from the original manuscript is irrelevant. The purpose of this review should be to better explain the literature, and to make some concepts clearer, so the Authors could put a bit more effort into their writing.
-
The claim that “Moreover, due to limitations in computational resources and accuracy, most studies have investigated gas-phase model systems of isolated chromophores or chromophore pairs” is totally misleading. Even in earlier studies, the protein environment was described at least approximately [ 10.1063/1.2210481 ]. As of today, virtually all calculations include the environment with QM/classical models.
Author Response
Reviewer 2
The Authors improved their review by addressing several of my points. They changed the title in order to remove the reference to computational studies. They generally improved the organization and readability of the review. However, I still think that the review is somewhat unbalanced. I do understand that it is not possible to cover all aspects of light-harvesting complexes in a short review. However, the Authors should strive to give a more balanced view of the literature (I) in presenting the experimental techniques, and (ii) when there is some debate related to the characteristics or function of the LHCs. For this reason, I believe that the manuscript should be further revised.
We would like to thank the reviewer once again for the constructive critique which hopefully has further improved our manuscript.
I understand that the Authors do not want to present all the experimental techniques used to understand the structure/function of LHCs. And I agree that the interpretation of 2DES is complex. However, the Authors should at least refer to the most recent review articles on the spectroscopic methods.
A short para and references to the most recent review articles on 2DES have been added.
Similarly, the Authors should acknowledge that MD simulations are now a fundamental tool to understand the structure and dynamics of LHCs, or to explore complex-complex interactions, even though they obviously are not perfect. The reasoning on the limitations of MD simulations is anyway specious, because (a) not all LHCs are membrane complex or contain carotenoids, (b) there exist ad-hoc force fields for carotenoid molecules, as well as Chls, (c) MD simulations can serve as a way to explore the conformational dynamics of LHCs, even if the force fields for the chromophores are not perfect. Also I do not understand the reference to the work by Cui and co-workers: the group actually uses MD simulations to understand the role of water in the conformational dynamics of membrane proteins.
The authors do not disagree with force field approaches being used for MD simulations - there is currently simply no alternative. We have made this more clear in the manuscript. Raised points:
(a) Chl description is also not perfect in force fields, especially the Mg/axial-ligand bond. The heme protein community has long resorted to including the metal/axial-ligand bond in the corresponding force field parameters; this is not the case for all Chl studies the authors are aware of.
(b) Force fields simply must fail for carotenoids unless they are tightly restrained to the structure they are parametrized for. We have shown for example in 10.1002/cphc.201402233 that the bond situation for fully symmetric xanthophylls is not centrosymmetric, but already broken by the protein environment.
(c) This was never disputed.
The work by Cui was mentioned to illustrate that problems occur at any triphasic interface, i.e., when water, membranes and proteins interact. With the new version of the manuscript this is not be relevant anymore.
I do not agree with the Authors on the delocalization length of the B850 ring. They deem all experimental results as unreliable, except for the one they chose. The latter one was determined from the enhancement of the transition dipole moment and comparison with very simple (exciton-only) modelling. Such combination of experiment and theory cannot distinguish a (pure) exciton delocalized state from a statistical mixing of states, and moreover it cannot separate the k=1 and k=-1 states (only one S0→S1 band is seen in the experiments). Therefore, these experiments only see the combined oscillator strength from more than one excited state. Nonetheless, experiments with several other techniques (in absorption, not in fluorescence) have measured a much shorter delocalization length [10.1021/jp9844415 10.1021/jp953639b ] and show that static disorder mixes the states of the perfect ring [ 10.1016/j.bpj.2014.03.023 ].
We obviously do not agree with the reviewer in this regard, the quoted refs. are not entirely convincing...
Another issue is on the OCP mechanism. The Authors are ignoring all the recent studies that support the translocation mechanism [10.1126/science.aaa7234 10.1016/j.tplants.2019.09.013 10.1073/pnas.1512240112 10.1021/jacs.7b05193 10.1038/s41598-017-15520-4 10.1021/jacs.8b11373l 10.1038/s41598-020-68463-8 ], thus giving a biased view of the literature on OCP. As of today, the literature agrees on the translocation mechanism. Note that there exist CTD/NTD homologous that are able to exchange bound carotenoids, which serves as an additional indirect proof of the translocation.
We agree with the reviewer in this regard, and have amended the respective sentences/refs. in the revised manuscript. However, please note - this is not a review on OCP nor NPQ mechanism(s).
If the Authors have a reference on the fact that “β-carotene in plant LHCI might be an isolation artifact”, they should cite it; otherwise, they should correct the sentence “In plants and algae, LHCs bind exclusively xanthophylls”
We do not agree with the reviewer at all; we think that carotenoids may be readily translocated by detergent treatment during isolation/purification of LHCI (and other complexes) – this is based on own experimental experience. Moreover, we are not aware of any convincing evidence/refs. for the reviewers’ claim.
I understand that this is not a review on NPQ, but the Authors should try to be more objective in their writing. Firstly, as I already pointed out, ref [139] does not use the correct QM tools to describe the S1 state of carotenoids. In addition, the fact that “S1 is never energetically below Qy unless it is in a S2 relaxed geometry [...]” is completely irrelevant. What is important is that the adiabatic S1 energy is lower than the Qy one, in which case the driving force for EET is negative, and the energy transfer is thermodynamically favoured by virtue of detailed balance considerations. Computational methods aside, the Authors should acknowledge that the spectroscopic literature (see my previous report) does support the population of a Car S1 state during quenching.
In principle we agree with the reviewer. With regard to the mentioned reference, we have deleted it. The reviewer also has a point regarding the adiabatic S1 energy to need to be below the Qy state for the mechanism to work.
In my opinion, it is important to mention that the isolated antenna complexes can switch between several conformations. There is independent evidence of this from single-molecule spectroscopy and transient absorption. This fact implies that the crystallographic view of LHCs is probably limited to a single structure in an ensemble of conformations.
We agree with the reviewer. This issue was raised already in a FEBS Lett. paper by Leupold et al. in 2000. Although for LH2…However, the reviewer should be aware that the environment of LHCs (in particular, LHCII) in single-molecule spectroscopy experiments is rather unnatural, to say at least …
The sentence “However, data obtained with NLPF on trimeric and aggregated LHCII seem to corroborate a Chl-xanthophyll excitonic interaction origin of NPQ [41].” seems to imply that there is an excitonic mixing of Qy and S1. Based on the Authors’ resonse to my previous report, it should be corrected by noting that the NLPF experiments show a change of interaction between Chls and Cars, but not necessarily an exciton state.
We fully agree with the reviewer in this regard, and have amended the respective sentence in the revised manuscript.
?? The sentence “PG is assumed in binding a Chl a which is supposed to be involved in EET to CP29 […]” should be rephrased because it is not clear. The fact that it was taken from the original manuscript is irrelevant. The purpose of this review should be to better explain the literature, and to make some concepts clearer, so the Authors could put a bit more effort into their writing.
We changed the sentence in the new version of the manuscript.
The claim that “Moreover, due to limitations in computational resources and accuracy, most studies have investigated gas-phase model systems of isolated chromophores or chromophore pairs” is totally misleading. Even in earlier studies, the protein environment was described at least approximately [ 10.1063/1.2210481 ]. As of today, virtually all calculations include the environment with QM/classical models.
Of course, studies which explicitly include the environment exist. But these studies are vastly outnumbered by the studies that consider isolated and/or individual Chls or carotenoids. For example, the work by Ch. Marian on carotenoids is exclusively in the gas phase (as they use them to calibrate their QM approaches). Also, the reviewer needs to carefully distinguish between studies that use the environment for ensemble generation (on the force field level) and then discard this information as it becomes too costly to include. This is the basis of most coupling approaches, which includes only single chromophores and then employs methods to construct the excitons (see for example several articles by Kühn, e.g. 10.1016/j.chemphys.2016.03.021). Please, note, however, that we have changed the sentence in the paper already for this revision to respond to a previous comment by the reviewer.